# Decision Tree Ensembles Utilizing Multivariate Splits Are Effective at Investigating Beta Diversity in Medically Relevant 16S Amplicon Sequencing Data

Josip Rudar,[a] G. Brian Golding,[b] Stefan C. Kremer,[c] Mehrdad Hajibabaei[a]

[a]Department of Integrative Biology & Centre for Biodiversity Genomics, University of Guelph, Guelph, Ontario, Canada
[b]Department of Biology, McMaster University, Hamilton, Ontario, Canada
[c]School of Computer Science, University of Guelph, Guelph, Ontario, Canada

**ABSTRACT**  Developing an understanding of how microbial communities vary across conditions is an important analytical step. We used 16S rRNA data isolated from human stool samples to investigate whether learned dissimilarities, such as those produced using unsupervised decision tree ensembles, can be used to improve the analysis of the composition of bacterial communities in patients suffering from Crohn's disease and adenomas/colorectal cancers. We also introduce a workflow capable of learning dissimilarities, projecting them into a lower dimensional space, and identifying features that impact the location of samples in the projections. For example, when used with the centered log ratio transformation, our new workflow (TreeOrdination) could identify differences in the microbial communities of Crohn's disease patients and healthy controls. Further investigation of our models elucidated the global impact amplicon sequence variants (ASVs) had on the locations of samples in the projected space and how each ASV impacted individual samples in this space. Furthermore, this approach can be used to integrate patient data easily into the model and results in models that generalize well to unseen data. Models employing multivariate splits can improve the analysis of complex high-throughput sequencing data sets because they are better able to learn about the underlying structure of the data set.

**IMPORTANCE**  There is an ever-increasing level of interest in accurately modeling and understanding the roles that commensal organisms play in human health and disease. We show that learned representations can be used to create informative ordinations. We also demonstrate that the application of modern model introspection algorithms can be used to investigate and quantify the impacts of taxa in these ordinations, and that the taxa identified by these approaches have been associated with immune-mediated inflammatory diseases and colorectal cancer.

**KEYWORDS**  16S rRNA, metric learning, amplicon sequencing, biomarker discovery, machine learning, metabarcoding, ordination

The analysis of the composition of ecological communities is essential to determine their function within the broader environment and within a host. The statistical analysis of differences between communities, beta diversity, is often an important aspect of statistical pipelines used to investigate ecological and metagenomic data sets. Central to the analysis of beta diversity is the calculation of pairwise distances, or dissimilarities, between samples. This calculation is carried out by carefully choosing and then applying a particular dissimilarity that is known to measure an important characteristic of the data. For example, some methods (such as correlation-based distance methods) perform well at measuring environmental gradients, while others, such as the Jaccard distance, can perform well when one is interested in clustering data (1–3). When analyzing amplicon sequencing data, another

**Copyright** © 2023 Rudar et al. This is an open-access article distributed under the terms of the Creative Commons Attribution 4.0 International license.
Address correspondence to Josip Rudar, joe.rudar@gmail.com, or Mehrdad Hajibabaei, mhajibab@uoguelph.ca.
The authors declare no conflict of interest.

typical goal is to discover amplicon sequence variants (ASVs) or operational taxonomic units (OTUs) associated with each type of community. This task is inextricably linked to variations in the composition of communities, which can be captured using a distance metric or dissimilarity measure. This type of analysis is typically carried out separately from the analysis of community composition using tools such as DESeq2, MetagenomeSeq, ANCOM, or linear discriminant analysis effect size (LEfSE) (3–8). Furthermore, the assumptions of these methods could contribute to higher false-positive and false-negative rates, which could lead to potentially erroneous or incomplete interpretations (3, 9, 10). Therefore, it is preferable if impactful ASVs (or OTUs, etc.) can be identified directly when investigating community composition. Furthermore, it is also important that these analyses are done using methods that make few (if any) assumptions about the underlying distribution of each ASV. Finally, for any method to be useful, it is important for it to consider dependencies between features (which can be genes, taxonomic groups, operational taxonomic units, or amplicon sequence variants). By attempting to model these dependencies, subtle differences between groups are more likely to be detected. This is supported by recent work that has demonstrated that genomic, transcriptomic, and metagenomic data sets are better understood if these relationships are considered (11–13).

Machine learning algorithms are uniquely suited to address these challenges. Unlike statistical models, machine learning models tend not to assume anything about the underlying distribution of each feature (4, 5). Furthermore, some machine learning models, such as random forest (RF) and related classifiers, are capable of identifying dependencies between features without the need for the user to explicitly include these dependencies in the model (11, 14–17). One ability, arguably underused, inherent to this class of models is that they can be used in an "unsupervised" manner to learn a dissimilarity function (15, 18, 19). This is known as metric learning, and the learned dissimilarity function can be used to replace a more traditional method (such as the Jaccard distance or Bray-Curtis dissimilarity) when investigating beta diversity (17, 20). This approach is also advantageous since it learns to remove the influence of uninformative features (21). Unsupervised random forests have previously been used to discover similar cell populations in single-cell transcriptome sequencing (RNA-seq) data, identify different classes of renal cell carcinoma tumors, and study the underlying structure of a population using shared genetic variations (20, 22, 23). If this approach is applied to amplicon sequencing data, it may be possible to simultaneously visualize the differences between communities while also identifying which features contribute most to the placement of each community within the projected space (17, 20). However, random forests do suffer an important limitation: the decision trees used to construct the forest make axis-orthogonal cuts. Randomizing the selection of cut points at each node has been shown to help, since this results in the construction of better decision boundaries. However, improvements like these are not a solution, since axis-orthogonal cuts are still made. To solve this problem, trees that can learn oblique or nonlinear cuts should be used. Recent and historical work using decision tree classifiers demonstrates that these types of splits often result in learning a more appropriate representation of the data (11, 24).

Recently, we introduced a new algorithm, large-scale nonparametric discovery of markers (LANDMark), as an alternative to RFs (25). Like the RF, LANDMark is a tree-based approach that recursively partitions samples until a stopping criterion is met. Unlike RFs, LANDMark can use learned linear and nonlinear models to partition samples (25). In this study, we investigate LANDMark's ability to predict medically relevant outcomes, such as Crohn's disease and colorectal cancer, using amplicon sequence data sets. We also provide a workflow, TreeOrdination, which uses LANDMark to quantify differences in the makeup of microbial communities in an unsupervised manner. TreeOrdination and LANDMark's performance are investigated using a synthetic and randomized data set, a small case study, and a larger colorectal cancer data set (26–28). The code used to perform the analysis in this work can be found at https://github.com/jrudar/Unsupervised -Decision-Trees.

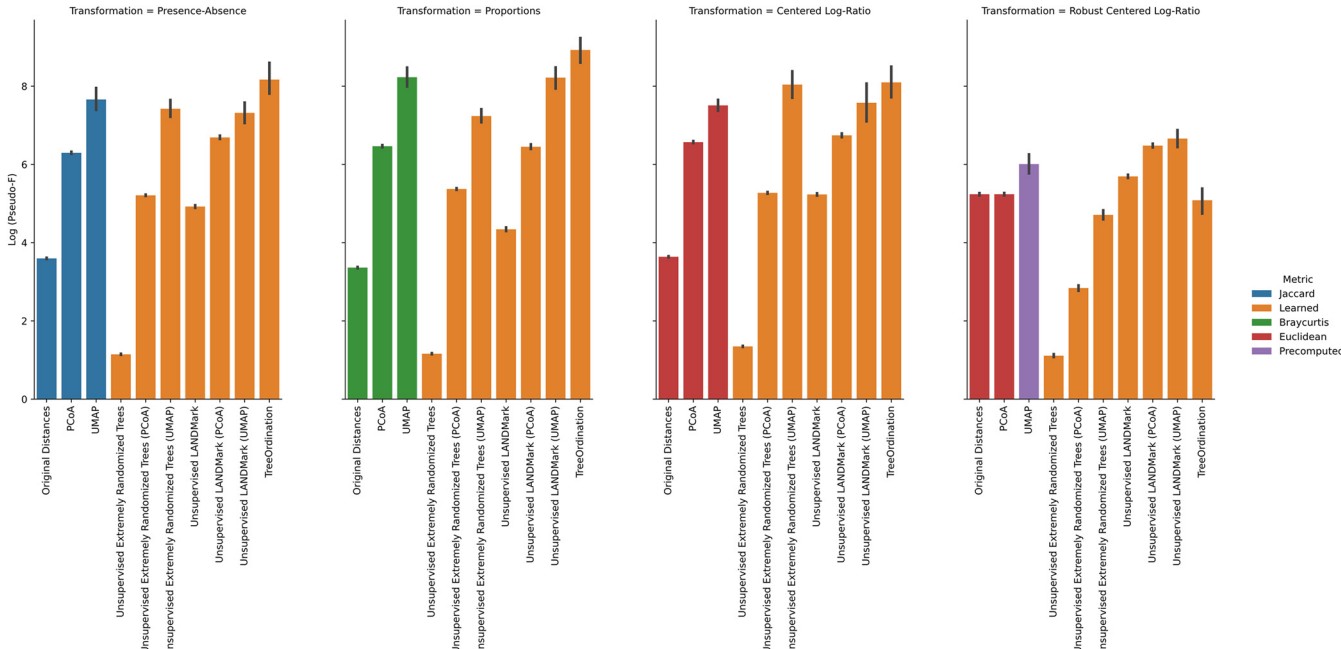

**FIG 1** PERMANOVA results for each test condition in the positive-control data. The pseudo-*F* statistic is log-transformed to improve the readability of the chart. Higher statistics indicate greater separation between groups of samples. The 95% confidence interval, calculated using 2,000 bootstraps, is shown for each bar in black. Five-fold stratified cross-validation with five repeats was used to generate these data. Bars are colored according to the dissimilarity. Blue, green, and red bars represent data analyzed using the Jaccard distance, Bray-Curtis dissimilarity, and Euclidean distances. Orange bars are learned dissimilarities, while purple bars are distances calculated using DEICODE.

## RESULTS

**TreeOrdination projections result in well-separated groups in simulated data.**
Under each transformation using the negative controls, we observed low *F* statistics and nonsignificant *P* values. This observation occurred irrespective of ordination type and model. Permutational multivariate analysis of variance (PERMANOVA) was able to identify significant differences ($P \leq 0.001$) between groups found in the positive controls across each cross-validation fold (Fig. 1), with Uniform Manifold Approximation and Projection (UMAP), unsupervised LANDMark, and TreeOrdination being among the best-performing methods (Fig. 1). However, the sole use of *F* statistics (or an equivalent statistic) to demonstrate class separation in the projection is not enough. Each ordination method should also be able to place new data of the same class into a similar location in the ordination space. When running this experiment, however, it quickly became apparent that creating ordinations using principal-coordinate analysis (PCoA) or robust principal component analysis (RPCA) resulted in an important limitation. Specifically, both methods were unable to transform and project new data into the space created using the training data. To do this, we would have needed to use all the available data (train and test), and this would have resulted in data leakage, since the test data would influence the final projection. Therefore, to prevent data leakage and the reporting of overly optimistic and potentially misleading results, the generalization performance of the RPCA and PCoA ordination methods was excluded (29). An overview of the important properties of each ordination method is presented in Table 1. Of the remaining transformations, the LANDMark, extremely randomized trees (Extra Trees), and TreeOrdination models were successful at classifying unseen samples (Fig. 2). Although statistically significant differences between models were observed, the size of these differences was likely small in this test, since all classifiers resulted in nearly perfect classification balanced accuracy scores (Fig. S2 in the supplemental material). In contrast, when training models using the negative-control data, we observed random performance (Fig. 3). Finally, our results showed that projecting data using UMAP resulted in good overall performance regardless of the distance or dissimilarity measure.

**TABLE 1** An overview of the properties of each ordination method

| Method | Can be trained using any ecological transformation | Can be used to transform unseen data | Learns/creates a nonlinear projection | Returns importance scores |
|---|---|---|---|---|
| TreeOrdination | Yes | Yes | Yes | Yes |
| Unsupervised LANDMark | Yes | Yes | Yes | Yes |
| Unsupervised extremely randomized trees | Yes | Yes | Yes[b] | Yes |
| Uniform manifold approximation and projection | Yes | Yes | Yes | No |
| Principal component analysis | No[a] | Yes | No | Yes |
| Principal coordinate analysis | Yes[a] | No | No | Yes |
| Robust principal component analysis (DEICODE) | No (robust centered log ratio only) | No | No | Yes |

[a]Assumes samples are found in a Euclidean space (PCA). PCoA is a generalization of PCA to other distance/dissimilarity measures.
[b]Although this is a type of nonlinear classifier, axis-orthogonal cuts are made to separate samples at each node.

**Learning dissimilarities can result in well-separated groups in the Crohn's disease case study.** In this study, we used the Crohn's disease data set as a follow-up real-world test. In this test, the centered log ratio (CLR) transformation consistently resulted in large *F* statistics and significant *P* values ($P \leq 0.05$) (Fig. 4). Except with CLR-transformed data, unsupervised extremely randomized trees underperformed and often resulted in ordinations where differences between Crohn's disease patients and healthy controls could not be consistently detected. Even in the best case (with the CLR transform), the *F* statistics associated with unsupervised extremely randomized trees ordinations were lower than those produced using the other ordination methods. We also observed that transforming data into proportions resulted in poor-quality ordinations. Although reasonable ordinations were created when using the Bray-Curtis dissimilarity, the *F* statistics associated with these ordinations were lower than those produced using the presence-absence and CLR transformations (Fig. 4). The cross-validated generalization performance of models trained on presence-absence and CLR-transformed data was the best (Fig. 5). No statistically significant difference was observed in the generalization performance between models trained on CLR-transformed data. However, statistically significant differences were observed between models trained on presence-absence data, proportions, and UMAP-transformed data (Fig. S3). The balanced accuracy scores calculated using TreeOrdination models were at least as good as those produced using LANDMark and Extra Trees. This result is surprising, because the predictive model in TreeOrdination is trained using a high-dimensional embedding learned by at least one unsupervised LANDMark classifier.

**LANDMark and TreeOrdination can be used with data sets where smaller effect sizes exist between conditions.** For models in this section, we will adopt the naming convention for models used in Baxter et al. (28). Specifically, classifiers trained on the combined ASV and fecal immunochemical test (FIT) data will be called ASV multitarget microbiota test (MMT) models. All models trained to distinguish between colorectal cancer and healthy controls performed better when incorporating FIT results with the microbiome

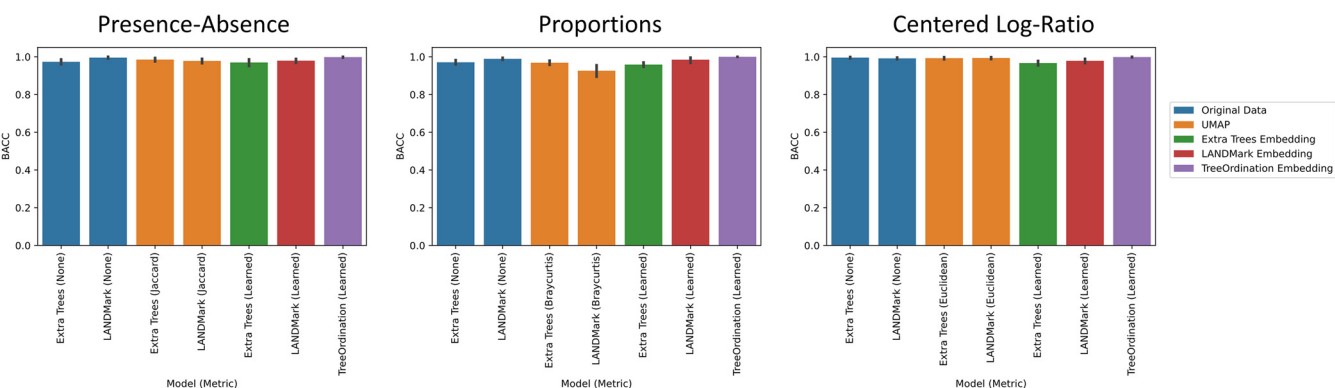

**FIG 2** Balanced accuracy score results for each test condition in the positive-control data. Higher scores indicate a more accurate classifier. The 95% confidence interval, calculated using 2,000 bootstraps, is shown for each bar in black. Five-fold stratified cross-validation with five repeats was used to generate these data.

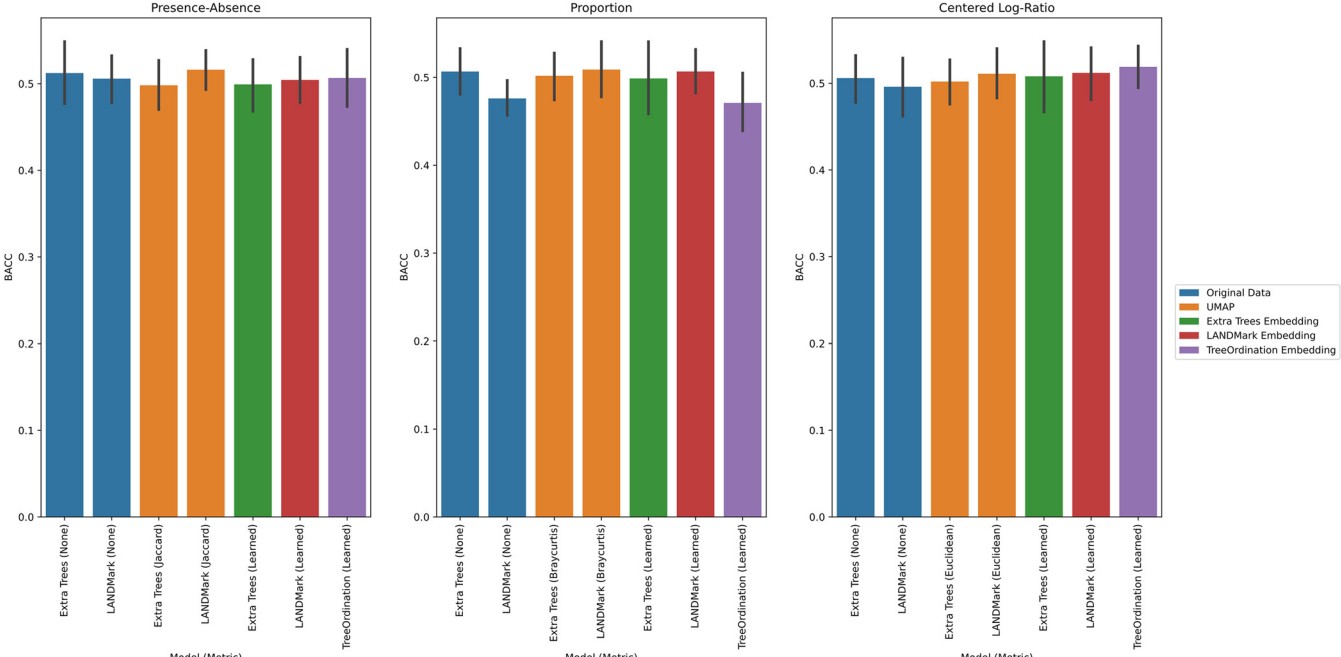

**FIG 3** Balanced accuracy score results for each test condition in the negative-control data. Higher scores indicate a more accurate classifier. The 95% confidence interval, calculated using 2,000 bootstraps, is shown for each bar in black. Five-fold stratified cross-validation with five repeats was used to generate this data.

data (Fig. 6). All models were better than random ($P \leq 0.0001$), and the LANDMark and TreeOrdination ASV MMT models performed particularly well in this test, with average receiver operating characteristic-area under the curve (ROC-AUC) scores for each model of 0.953 and 0.953, respectively. These performed significantly better than the Extra Trees and random forest classifiers ($P \leq 0.0001$), and the ROC-AUC score from both models is comparable to that from Baxter et al. (2016) (0.952) (28). However, using ROC-AUC scores as a measure of generalization performance can be optimistic when used with models trained on imbalanced data sets (such as this one). To address this problem, we used the balanced accuracy score, since it is a better measure with imbalanced data. We found that the scores produced by the LANDMark and TreeOrdination ASV MMT models (mean balanced accuracy score of 0.88) were better than those of the competing methods used here (Fig. 6) and that this difference was statistically significant ($P \leq 0.0001$). Since the TreeOrdination and LANDMark models performed best, we used them to calculate the discrimination thresholds that maximized the detection of colorectal cancer while minimizing the detection of healthy tissue. These thresholds were 0.30 for LANDMark and 0.43 for TreeOrdination. When using these thresholds, both ASV MMT models discovered a substantial fraction of additional colorectal cancers (Fig. 7 and Table 2), with the LANDMark model having a higher sensitivity when than FIT and TreeOrdination, at a cost of a lower specificity (Table 2).

While detecting colorectal cancer is important, it is also important to detect precancerous adenomas. Therefore, we trained a model to distinguish between lesions and healthy tissue. We found that statistically significant differences existed between the generalization performance of various models and that models that included FIT results performed better (Fig. 8). When training each classifier using the combined ASV and FIT score data, we observed that the balanced accuracy scores of random models did not differ significantly from those of random forests or Extra Trees. However, the LANDMark and TreeOrdination ASV MMT models both performed significantly better than Extra Trees ($P \leq 0.0001$) and random forest ($P \leq 0.0001$) classifiers trained on the same data. Both models resulted in a mean balanced accuracy score of 0.63. Once again, we calculated the optimal discrimination threshold for distinguishing lesions (adenomas and colorectal cancers) from healthy tissue. This threshold was found to be 0.69 for LANDMark and 0.66

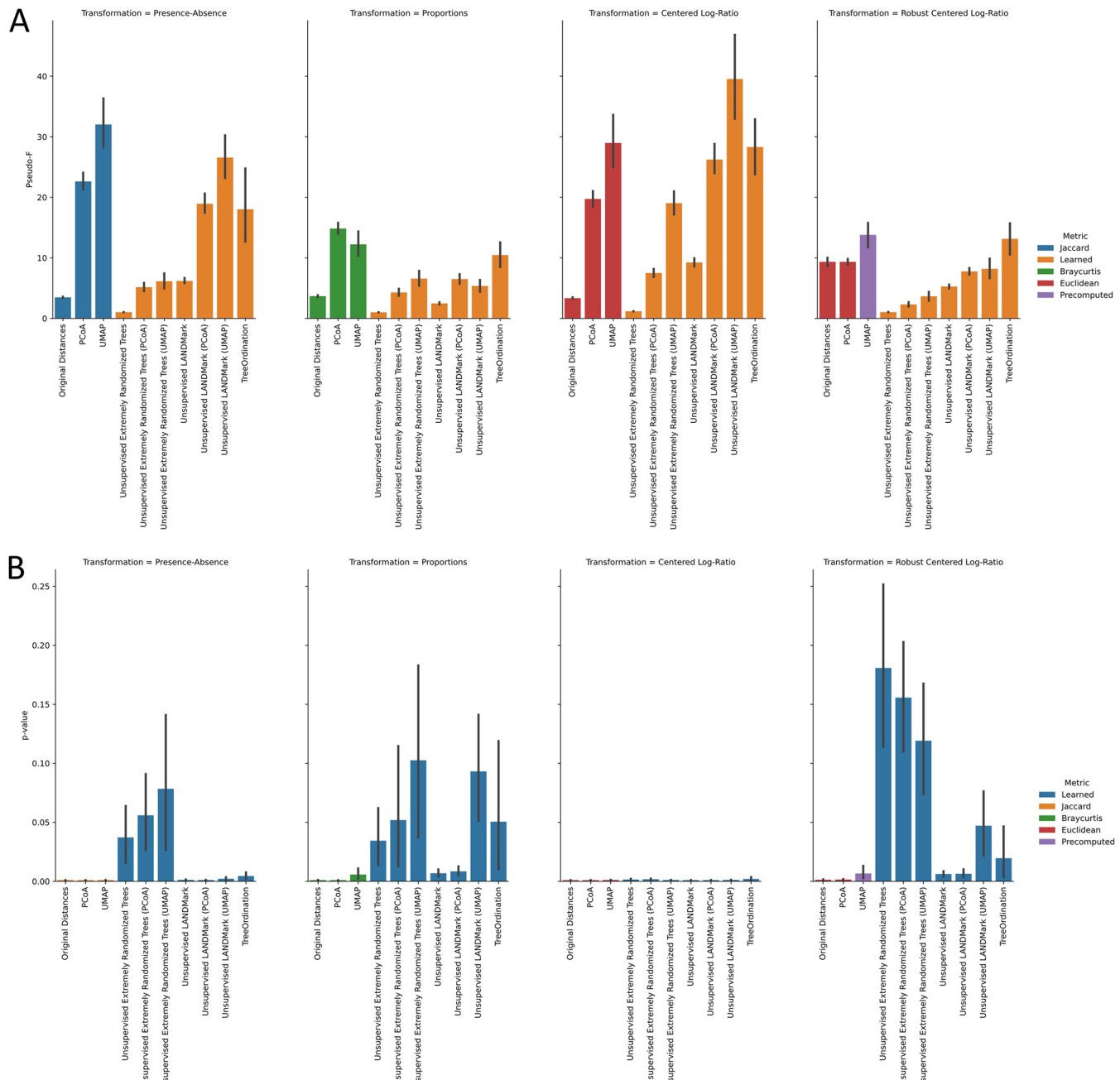

**FIG 4** PERMANOVA results for each test condition in the Crohn's disease data. (A) Higher pseudo-*F* statistics indicate greater separation between groups of samples. (B) Distribution of *P* values associated with the cross-validated distribution of pseudo-*F* statistics shown in panel A. The 95% confidence interval, calculated using 2,000 bootstraps, is shown for each bar in black. Five-fold stratified cross-validation with five repeats was used to generate these data.

for TreeOrdination. Compared to FIT alone, the ASV MMT models resulted in better detection of additional colorectal cancers and adenomas (Table 2). Interestingly, while the LANDMark and TreeOrdination models both performed well, they differed in their ability to identify different types of lesions, with TreeOrdination trading sensitivity for increased specificity and the opposite occurring in LANDMark (Fig. 9).

**TreeOrdination can be used to identify features that have a high impact on model performance.** In the colorectal cancer data set, the best discrimination threshold for a TreeOrdination ASV MMT model was found to be 0.67. While the ability of the model to identify lesions was lower than that of LANDMark (Table 2), this came with a marked improvement in specificity (Table 2 and Fig. 9). A TreeOrdination projection, using the

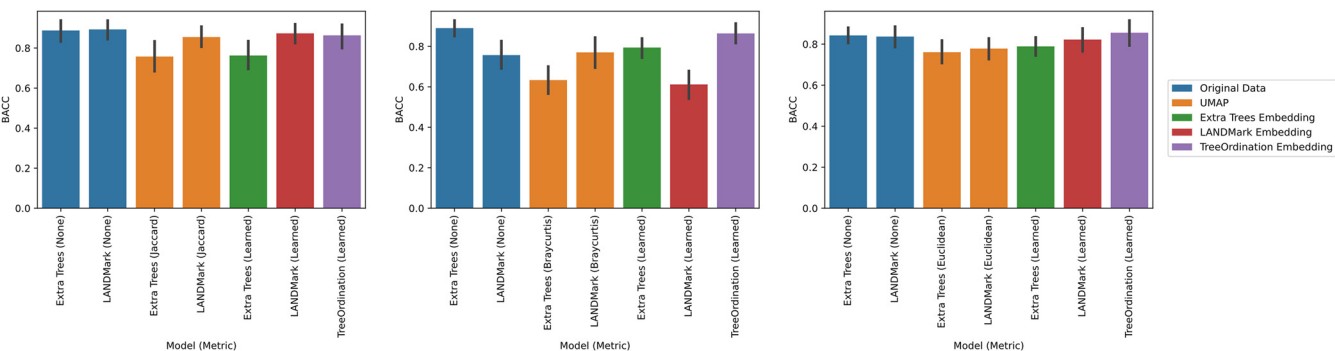

**FIG 5** Balanced accuracy score results for each test condition in the Crohn's disease data. Higher scores indicate a more accurate classifier. The 95% confidence interval, calculated using 2,000 bootstraps, is shown for each bar in black. Five-fold stratified cross-validation with five repeats was used to generate these data.

approximate embedding function, of predicted lesions (samples over the discrimination threshold) and healthy samples was created using the test set data (Fig. 10A). This transformation was a reasonable projection, since the mean squared error between this approximation and the full embedding was small (1.1). Clear differences in the locations of lesions and healthy tissue can be seen in the plot. This suggests that there are differences between patients with colorectal lesions and those with healthy tissue. This observation is supported by way of a significant PERMANOVA result (pseudo-$F$ = 375.12, $P \leq 0.001$, $R^2$ = 0.99). Most of the variation between samples, approximately 87%, can be explained by the first principal component, which appears to be associated with disease status. The variation along this component appears to be predominantly driven by FIT scores (Fig. 10B, 11, and 12). Statistically significant differences were detected in the projection of adenoma and carcinoma test samples (pseudo-$F$ = 9.51, $P \leq 0.002$, $R^2$ = 0.73), with variation between these groups being driven by FIT score (Fig. 10B). However, differences in the microbiome also appeared to play a role in determining where these samples were placed. While the impact of the microbiome was minor compared to that of the FIT scores, ASVs assigned

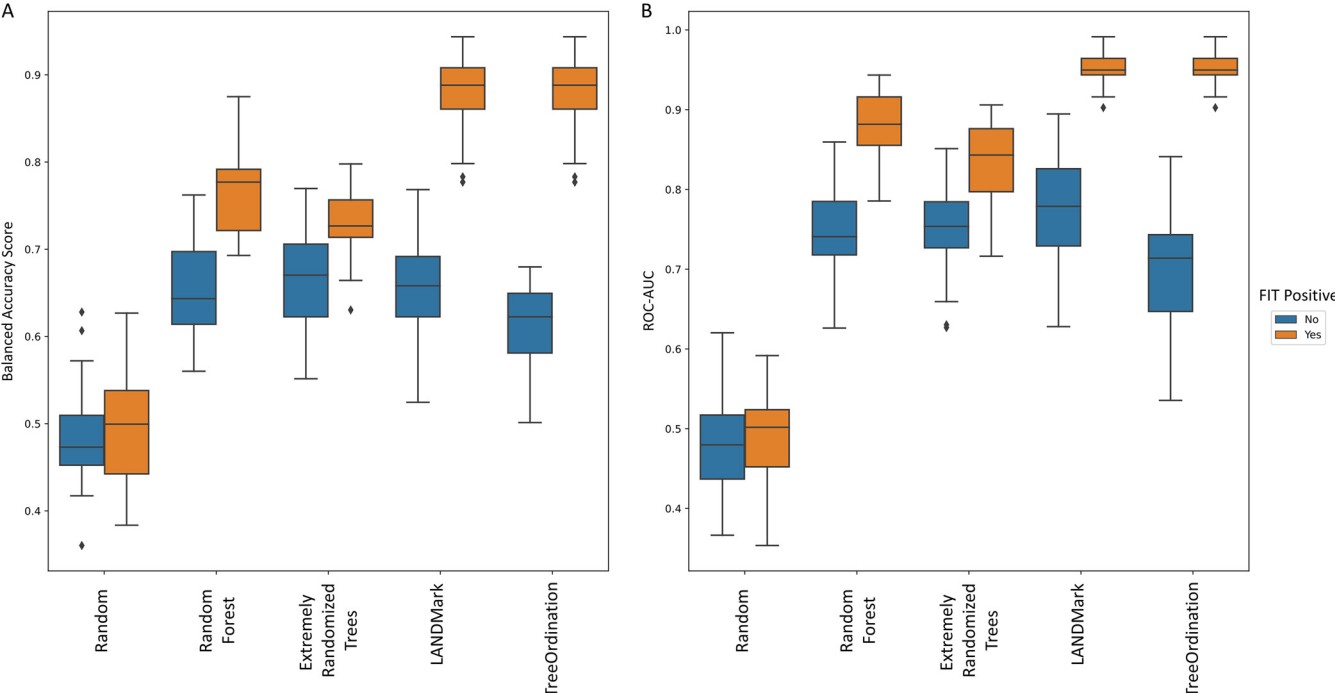

**FIG 6** The ability of each model to identify patients with colorectal cancer. (A) Balanced accuracy scores. (B) ROC-AUC scores. Each model was trained using centered log ratio-transformed data with (orange bars) and without (blue bars) the inclusion of FIT scores. These scores were generated using fivefold stratified cross validation with five repeats.

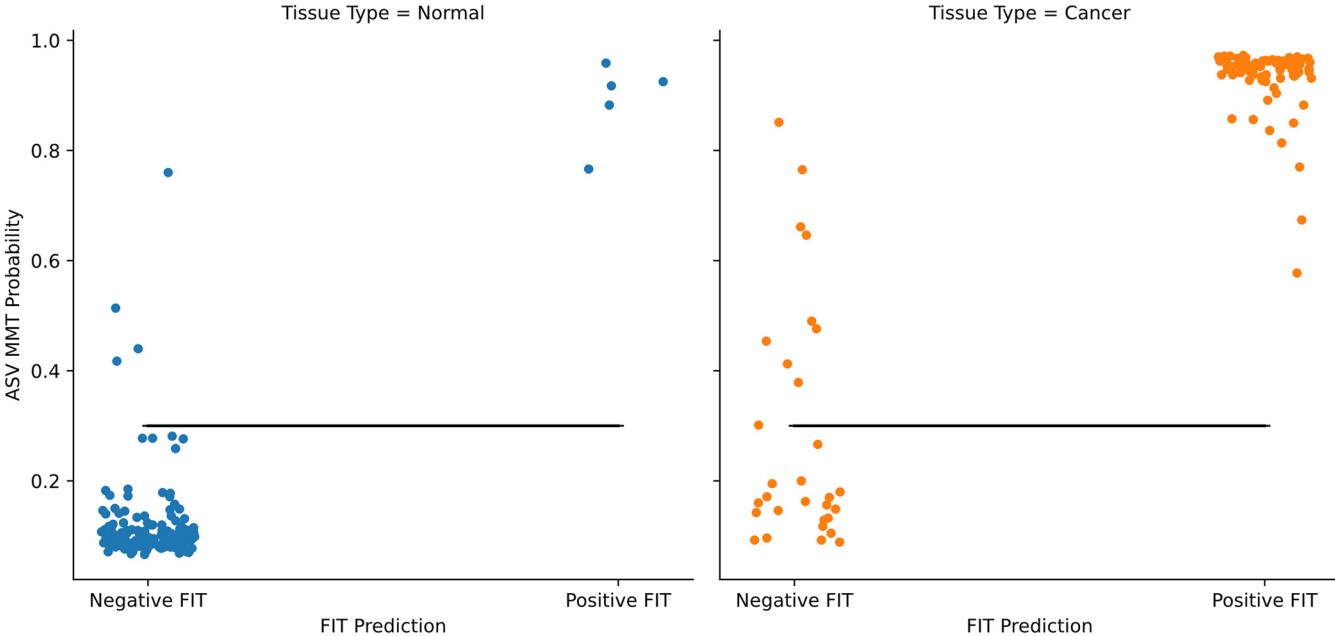

**FIG 7** Results of the LANDMark-based ASV MMT model for distinguishing between healthy tissue and colorectal cancer. Points are colored based on the predicted labels and actual labels. The positions of points along the *y* axis were found by averaging the cross-validated probabilities of test samples. The black horizontal lines represent the threshold maximizing the balanced accuracy score (0.30).

to *Lachnospiraceae*, *Clostridiales*, *Anaerotruncus*, and *Ruminococcaceae* were among the top features contributing to the placement of adenomas and carcinomas. Although the microbiome appeared to play a more muted role, the sum of the Shapley scores for ASVs in FIT-negative lesions tended to carry greater weight than that of their FIT-positive counterparts. This could indicate that FIT scores were used to determine the region in which a sample is found, while the composition of the microbiome was used to refine the location of the sample within that region. Evidence for this comes from the higher Shapley scores assigned to ASVs in FIT-negative samples (Fig. 11 and 12).

The mean squared error between the full and approximate TreeOrdination projections created using the Crohn's disease data set was 0.54, indicating that the approximate embedding is reasonable. Differences between the stool microbiomes of Crohn's disease patients and healthy controls were detected using PERMANOVA (pseudo-$F = 31.06$, $P \leq 0.003$, $R^2 = 0.99$). Nearly all of the variation between samples, approximately 84%, could be explained by the first principal component. This strongly suggested that disease status was associated with the variation along this component. The ASVs that were

**TABLE 2** Sensitivities and specificities for LANDMark and TreeOrdination ASV MMT models, TreeOrdination ASV MMT models, and FIT

| Model | Mean value (95% CI)[a] | |
| --- | --- | --- |
| | Sensitivity | Specificity |
| Healthy vs cancer (FIT) | 0.75 (0.68–0.83) | 0.97 (0.94–0.99) |
| Healthy vs adenoma (FIT) | 0.16 (0.11–0.21) | 0.97 (0.94–0.99) |
| Healthy vs cancer only (TreeOrdination MMT model) | 0.82 (0.74–0.88) | 0.96 (0.94–0.99) |
| Healthy vs cancer only (LANDMark MMT model) | 0.83 (0.76–0.90) | 0.95 (0.91–0.98) |
| Healthy vs cancer (lesion TreeOrdination MMT model) | 0.83 (0.77–0.90) | 0.96 (0.96–0.98) |
| Healthy vs adenoma (lesion TreeOrdination MMT model) | 0.25 (0.19–0.31) | 0.96 (0.93–0.98) |
| Healthy vs lesion (lesion TreeOrdination MMT model) | 0.47 (0.43–0.52) | 0.96 (0.94–0.99) |
| Healthy vs cancer (lesion LANDMark MMT model) | 0.89 (0.83–0.94) | 0.86 (0.80–0.91) |
| Healthy vs adenoma (lesion LANDMark MMT model) | 0.36 (0.30–0.43) | 0.86 (0.81–0.91) |
| Healthy vs lesion (lesion LANDMark MMT model) | 0.56 (0.52–0.61) | 0.86 (0.81–0.91) |

[a]Each 95% confidence interval was calculated using 2,000 stratified bootstrap replicates.

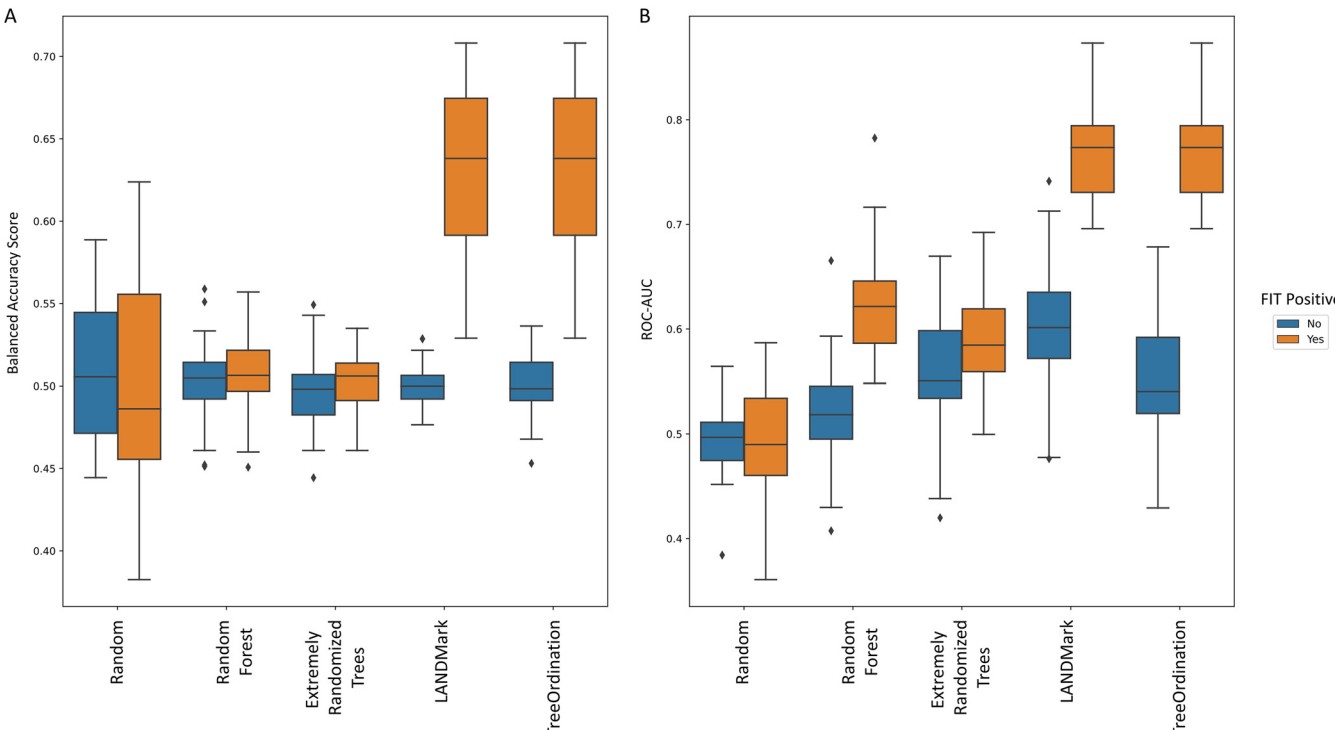

**FIG 8** The ability of each model to identify patients with colorectal lesions (adenomas and colorectal cancers). (A) Balanced accuracy scores. (B) ROC-AUC scores. Each model was trained using centered log ratio-transformed data with (orange bars) and without (blue bars) the inclusion of FIT scores. These scores were generated using 5-fold stratified cross-validation with five repeats. LANDMark and TreeOrdination models trained on the combined ASV and FIT data tended to perform better than their counterparts in this task.

predominantly associated with this variation belonged to *Fournierella* spp., *Staphylococcus* spp., *Dialister* spp., and *Ruminococcaceae* (Fig. 13).

## DISCUSSION

The data sets investigated here were chosen because the human gut microbiome is an important area of medical research and is becoming increasingly linked to important disease phenotypes (26, 28, 30, 31). Since machine learning models are becoming increasingly used to identify predictive features, it is important to develop models that integrate ordination, prediction, and the detection of differentially abundant taxa so that a better understanding of the system and the composition and function of the human microbiome can be developed. The choice of transformation and dissimilarity measure is often an important first consideration when investigating microbiome data. It has long been known that the choice of dissimilarity measure can influence our measurement and interpretation of the main gradients influencing the structure of communities and the taxonomic similarity between pairs of samples (32, 33). For example, recent investigations have demonstrated that this choice can result in misleading results due to the sparsity inherent in the data and to differences in library size and sampling (34–36). To combat these problems, a multitude of dissimilarity measures and ordination approaches have been developed to summarize and visualize ASV differences between sites (33). However, distance metrics and other commonly used dissimilarity measures have difficulty capturing potential dependencies between ASVs. For example, the Jaccard distance simply calculates the number of shared ASVs over the total number of unique ASVs between two communities, and it fails to consider whether the presence or absence of one ASV influences another (16). Furthermore, when using measures that use abundance information, differences in abundance can create situations where the sites that share the same species are more dissimilar than sites that have no species in common. Finally, as the dimensionality of the data set

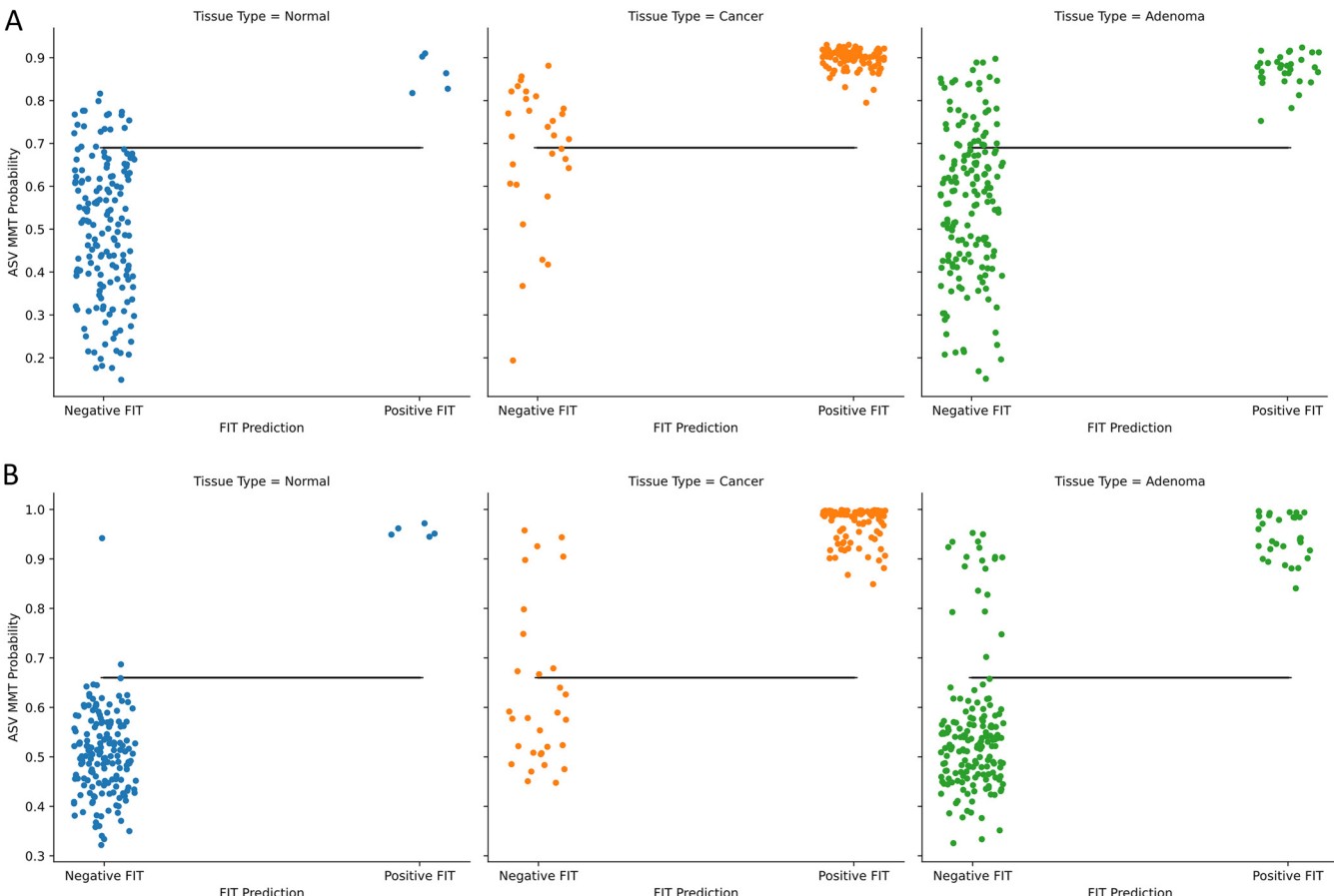

**FIG 9** The discriminatory power of LANDMark and TreeOrdination-based ASV MMT models. (A) The ability of LANDMark to distinguish between healthy tissue, adenomas, and colorectal cancers at the given discrimination threshold (0.69). (B) The ability of TreeOrdination to distinguish between healthy tissue, adenomas, and colorectal cancers at the given discrimination threshold (0.66). The positions of point along the *y* axis were found by averaging the cross-validated probabilities of test samples. The black lines represent the threshold maximizing the balanced accuracy score (0.69).

grows (the number of OTUs or ASVs increases), there is an increasing probability that many of these dimensions add noise to the data set (37, 38). This can occur for several reasons. For example, chimeric reads, rare sequences, inefficiencies during amplification due to primer design, and bioinformatic pipeline can contribute irrelevant information that is difficult to remove (36, 39). Many contemporary dissimilarity measures are often sensitive to this noise, since they have no way of reducing it when the dissimilarity is being calculated (37, 38). Although the application of transformations, such as the centered log ratio or converting to presence-absence, can mitigate some of these issues, there is yet to be a consensus on which approach is best (3, 33, 34, 40).

It is also important to address the potential weaknesses of ordination methods like PCoA, robust principal component analysis (RPCA) (27), and UMAP. To begin, both PCoA and RPCA are unable to transform new data. This is a problem since testing and validation sets are unable to be transformed using these methods without introducing data leakage. Data leakage, in this case, can occur if projections are created using the concatenated data set (all training, testing, and validation samples are used). When this happens, information about any withheld data are included when the PCoA or RPCA objective function is optimized. This could bias the results and potentially create machine learning models that produce overly optimistic and potentially misleading results (29). With UMAP, this is not a problem, since UMAP can learn an appropriate transformation using only the training data. This transformation can then be applied to test data or new data without data leakage occurring. PCoA, RPCA, and to a lesser extent, UMAP are also problematic in that they require a dissimilarity measure to create an ordination. This potentially makes these methods

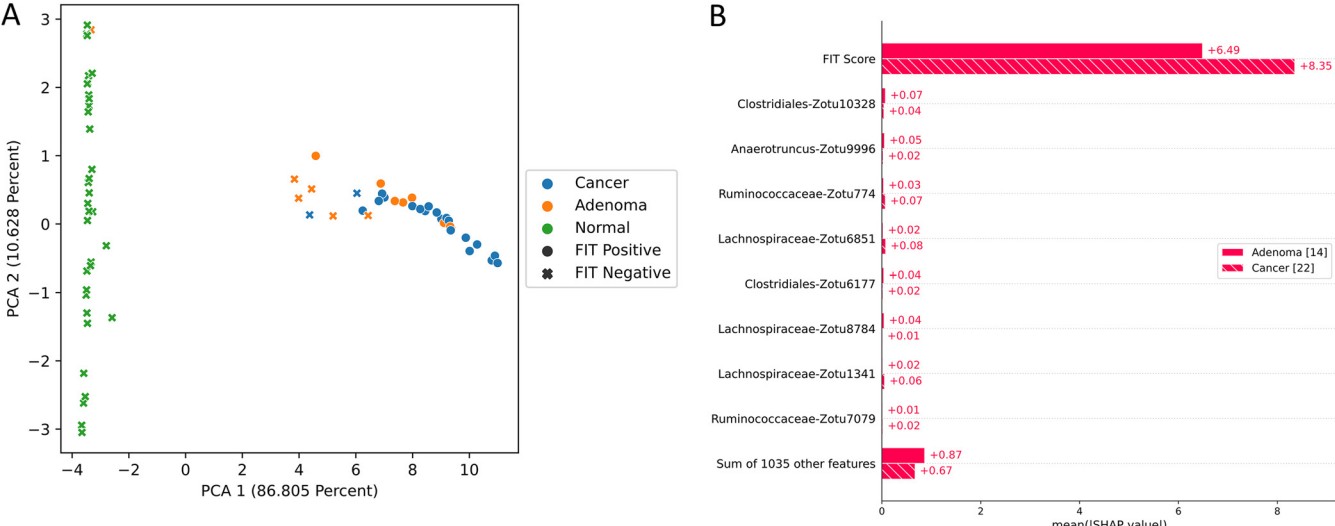

**FIG 10** TreeOrdination projection of test samples. (A) PCA axis 1 versus axis 2 constructed using the subset of lesions (colorectal cancers and adenomas) above the discrimination threshold and all healthy samples below this threshold. (B) A summary of how the Shapley values impact the placement of adenomas and carcinomas along the first principal component.

susceptible to the previously discussed assumptions and issues that come with the use of such measures.

Our work presents results that attempt to address a gap ecology and microbial ecology by using metric learning to model and understand community structure. These approaches to measuring pairwise dissimilarity have been applied to the analysis of genomic and transcriptomic data sets (18, 20, 22, 41). Our foray into this domain shows that learning the pairwise dissimilarities between samples can result in good projections that preserve the suspected biological differences between groups. We show that UMAP is an excellent out-of-the-box replacement for PCoA and RPCA. When using UMAP projections, we observed larger separations between groups (measured using PERMANOVA) and models trained on the projected data were accurate at identifying the class of test samples (Fig. 4 to 6). Furthermore, training models using these representations resulted in generalization performance that was at least as good as that of competing methods (Fig. 1 and 2). This supports a growing body of work that advocates for using UMAP to study community composition (13, 42). We also attempt to address the problems associated with contemporary measures of dissimilarity (15, 16, 43). Specifically, we introduce an approach, TreeOrdination, that can learn the pairwise dissimilarities between samples while potentially accounting for dependencies between features, since it uses a classifier, LANDMark, that makes multivariate splits at each node. In previous work, we have shown that ensembles built using such splits do a better job of modeling the underlying patterns in data (25). This is also well supported by other work in the field (15, 17, 24, 41). By using LANDMark, TreeOrdination also minimizes the impact of noisy features through randomization (bootstrapping of training data at each node, random selection of features, and models) and regularization (most models selected for splitting are L1 or L2 regularized) (21, 25). TreeOrdination also mitigates the weaknesses of PCoA and RPCA because it can transform new data after training. Furthermore, it does not require a distance metric, since the dissimilarity function is learned. TreeOrdination can do this because it first creates an intermediate embedding of the training data before using this embedding as input for a UMAP projection (18, 19, 22, 25). Once the initial embedding is learned, the training data are no longer needed, since each LANDMark model has learned to create an internal representation of how to partition the space upon which the training samples are found (24, 25, 41, 43). The locations of testing samples in this space can then be extracted and transformed into a lower-dimensional space using UMAP. We show that this process works well when applied to real metagenomic data sets. In these data sets, TreeOrdination's performance in these data sets was competitive with that of its peers. However, we must note that

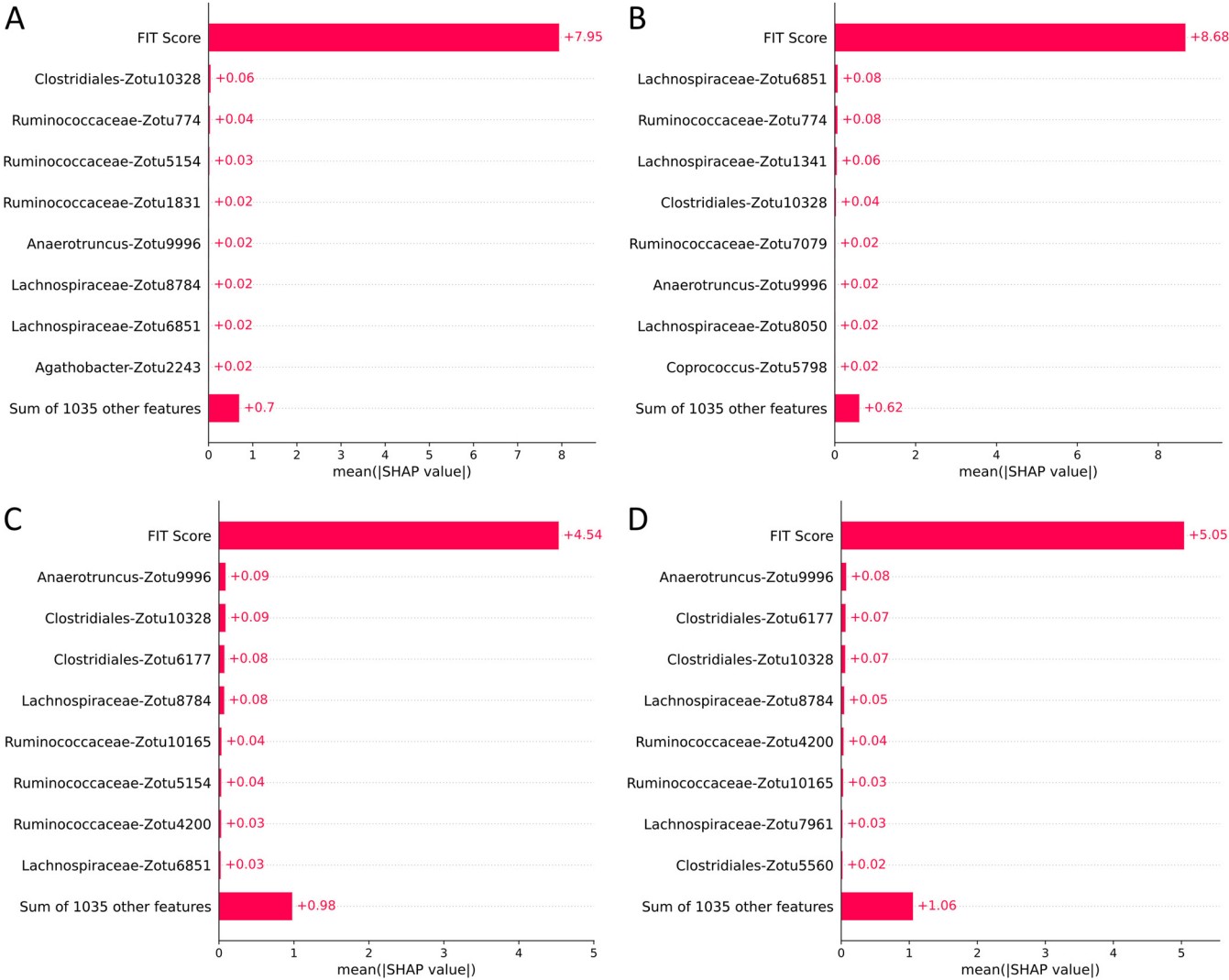

**FIG 11** Global feature impact scores in FIT-positive and FIT-negative samples. Impact of each feature's ability to discriminate the locations of samples along PC1 in the TreeOrdination projection of FIT-positive adenomas (A), FIT-positive colorectal cancers (B), FIT-negative adenomas (C), and FIT-negative colorectal cancers (D).

we only examined two data sets and metric learning approaches may not be suitable for all data. This is an important limitation of this study, and therefore, it is up to the investigator to explore potential models and use the one best suited for the data and question being asked.

A final contribution of our work is the link we identify between the original high-dimensional representation of the data and the lower-dimensional projection. Specifically, we use Shapley values to understand the impact that each feature (ASVs, OTUs, etc.) has on the ordination. In ordination approaches like PCoA, this information is exposed as loading scores along each principal component. However, these methods are limited in their ability to project data when the pattern separating groups of samples is nonlinear. Furthermore, they are sensitive to outliers due to their reliance on a distance measure (37). Assuming groups can be separated using some nonlinear function, the weakness of these approaches can be easily verified if low PERMANOVA $F$ statistics and high reconstruction losses between distances in the ordination and the original space are observed. While UMAP represents an important improvement, since it is a graph-based method that can model these nonlinearities, we lose access to the loading scores and interpretability if it is used (13, 44, 45). To address this problem, Shapley values can be used to peer through the "black box" and quantify the impact that each feature has on the projection (46, 47).

Our results support this approach, since we have identified ASVs belonging to organisms associated with inflammation and colorectal cancer. This will not be an exhaustive

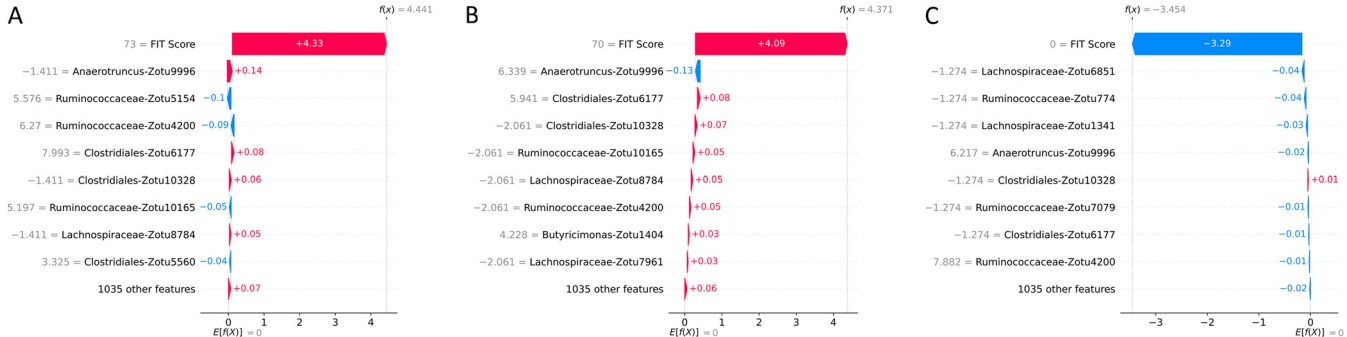

**FIG 12** Local importance values for individual adenoma, colorectal cancer, and healthy FIT-negative samples. The placement of individual samples along the first principal component [the *f(x)* value] can be decomposed using Shapley values. Examples of how ASVs and FIT scores impact the locations of an adenoma (A), a colorectal cancer (B), and a healthy (C) sample are shown. Along the left in gray is the FIT score and the abundance of each ASV relative to the geometric mean of the sample. Points are shifted to the left (blue) if they are pushed toward the "healthy" region of the ordination and to the right (pink) if they are pushed to the region concentrated with lesions.

discussion of the organisms identified, since the aim of this work is to show that methods like TreeOrdination are feasible. We observed that ASVs associated with *Ruminococcaceae* were important in refining the locations of samples in FIT-negative adenomas and colorectal cancers and determining the locations of samples in the Crohn's disease data set in TreeOrdination models. The identification of this taxon is important, since there is evidence to suggest that the depletion of this family is associated with colorectal cancer and gut inflammation (48, 49). Indeed, relative to the geometric mean, the depletion of *Ruminococcaceae* in the test samples pushes them to be more lesion-like (Fig. 12). Our results also found commonalities with those reported in the original study. For example, our results and those from Baxter et al. show that *Lachnospiraceae* is an important predictive component, and a low abundance of this group tends to push models to be more lesion-like (28, 49). *Anaerotruncus* also appeared to play a role in refining the locations of FIT-negative samples. The abundance of this genus relative to the geometric mean of each sample suggests that high abundance will push samples to be less lesion-like. This is an encouraging result, since work has shown that an elevated abundance of *Anaerotruncus*, a butyrate producer, may exert a protective effect by inhibiting glucose transport and glycolysis (50, 51). Additional support for our method comes from the results of the Crohn's disease data set. *Fournierella* spp., *Staphylococcus* spp., *Dialister* spp., and *Ruminococcaceae* were identified as the organisms with the strongest impact on the locations of samples in the

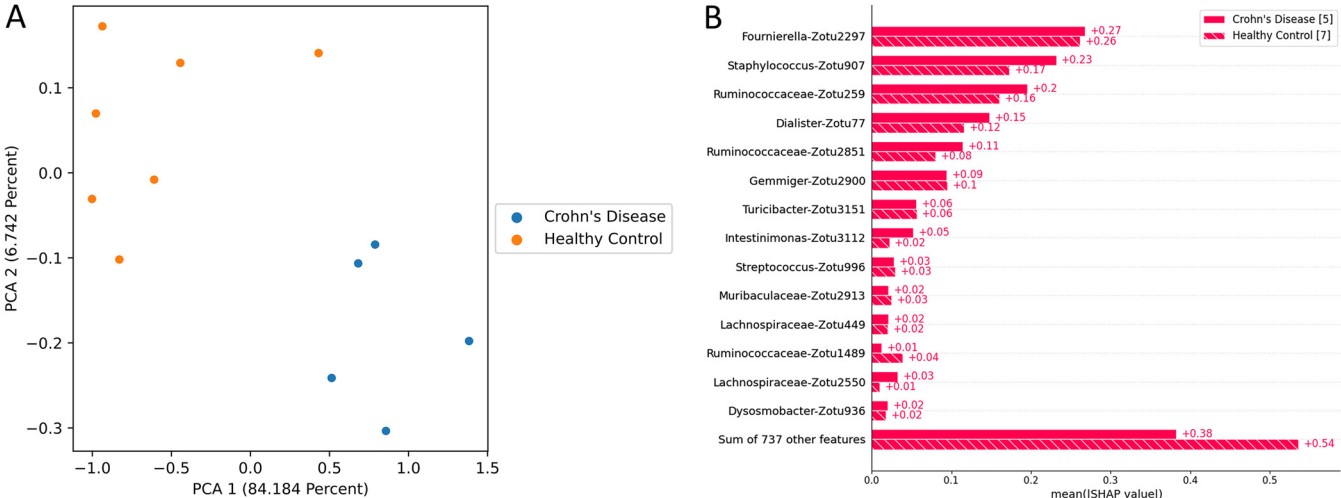

**FIG 13** TreeOrdination projection of test samples from the Crohn's disease data set. (A) PCA axis 1 versus axis 2 constructed using test samples from the Crohn's disease data set. (B) A summary of how the Shapley values impact the placement of samples along the first principal component.

TreeOrdination projection. We also identified ASVs assigned to *Ruminococcaceae*, *Gemmiger* spp., and *Lachnospiraceae* as those that impact the locations of samples in the TreeOrdination projection. Encouragingly, these were also detected as important in the original study of Forbes et al. (26). Some taxa, such as *Staphylococcus* and *Dialister*, have known associations with Crohn's disease, proinflammatory phenotypes, and other inflammatory diseases, such as ankylosing spondylitis (52–54). In addition, detecting ASVs assigned to *Ruminococcaceae* and *Lachnospiraceae* as impactful is an important validation, since these families are known to be beneficial due to being producers of anti-inflammatory metabolites (28, 55). We do acknowledge that our investigation differed from the original investigations due to our use of ASVs. However, our results are generally consistent with the wider interpretation of these studies: the composition of the microbiome tends to shift toward species that exert a proinflammatory effect (26, 28). Finally, these results demonstrate the power of TreeOrdination, since the factors contributing to the placement of individual samples can be investigated. This allows investigators to use a single model to investigate community composition and differential abundance and predict the properties of new samples.

**Conclusions and future work.** Our work has shown that unsupervised LANDMark (Oracle) and TreeOrdination models can learn effective dissimilarity matrices. When paired with modern dimensionality reduction approaches, such as UMAP, the global structure of the data set is preserved. These representations can then be combined with existing matrix factorization and model introspection algorithms to create informative ordinations. Additional work should be conducted on larger data sets to understand when metric learning should be used. Furthermore, by using larger data sets, the effect of individual differences will be reduced, since a larger sample of the population will be used in each test set (30). This will also be a good opportunity to conduct additional comparisons between other commonly used transforms, dissimilarity measures, and read-clustering approaches. Finally, with respect to TreeOrdination, additional work needs to be conducted in order to identify significantly different features. While there is still work to do, our results support the use of metric learning as a replacement for traditional dissimilarity measures in ecological research. These are advantageous since they can be used to limit the influence of outliers and uninformative features. In the data sets used here, we show that there is a shift toward a proinflammatory microbial community in the stool samples of patients suffering from Crohn's disease and colorectal cancer and that our results are similar to those found elsewhere in the literature. Finally, additional work should be conducted to determine more optimal ways of calculating the dissimilarity between samples when using decision tree ensembles. This will hopefully lead to improved ordinations and interpretations.

## MATERIALS AND METHODS

**Data set description.** Two human microbiome data sets were selected for use as case studies. The first data set was chosen because it contained samples from patients who suffered from immune-mediated inflammatory diseases (IMID) (26). The microbiomes of patients suffering from Crohn's disease, ulcerative colitis, multiple sclerosis, and rheumatoid arthritis were compared to those of healthy controls. This is an important area of research, since many people across the world suffer from these conditions. Therefore, models that are better able to identify ASVs associated with these conditions can provide further insight into these diseases. In this study, we used the Crohn's disease-healthy control subset of data as a case study. The second data set was derived from the stool samples of healthy patients and those suffering from colorectal cancers and adenomas. 16S rRNA gene sequencing was used to identify the composition of the colon community in each of the 490 patients (28). This data set is important due to the prevalence of colorectal cancers in our society. Therefore, the ability to train models to detect the early stages of cancer development and colorectal cancers will result in economic savings and the saving of lives.

**Bioinformatic processing of raw reads.** Raw sequences for each of the data sets were obtained from the Sequence Read Archive (accession numbers PRJNA450340 and SRP062005) (26, 28). All bioinformatic processing of the raw reads was prepared using the MetaWorks version 1.8.0 pipeline (https://github.com/terrimporter/MetaWorks) (56). The default settings for merging reads were used except for the parameter controlling the minimum fraction of matching bases, which was increased from 0.90 to 0.95. This was done to remove a larger fraction of potentially erroneous reads. Merged reads were then trimmed using the default settings that MetaWorks passes to CutAdapt. Since reads from the record with accession number SRP062005 were preprocessed and the primers removed, no reads were discarded during trimming. The remaining quality-controlled sequences were then dereplicated and denoised using VSEARCH version 2.15.2 to remove putative chimeric sequences (57). Finally, VSEARCH was used to construct a matrix where each row was a sample and each column an amplicon sequence variant (ASV). Taxonomic assignment was conducted using the RDP

Classifier (version 2.13) and the built-in reference set (58). ASVs that were likely to be contaminants, specifically those likely belonging to chloroplasts and mitochondria, were removed. From the remaining sequences, only those belonging to the domains *Bacteria* and *Archaea* were retained for further analysis. A sample-by-ASV count matrix, where each row was a sample and each column an ASV, was created for each data set.

**Calculation of dissimilarities.** Pairwise dissimilarities between samples were calculated using the Jaccard distance for presence-absence data (equation 1), the Aitchison distance for centered log ratio and robust centered log ratio data (equation 2), and the Bray-Curtis dissimilarity (equation 3) for data converted into proportions. In addition, we investigated the performance of extremely randomized trees (Extra Trees) dissimilarity (Equations 4 and 5) and LANDMark dissimilarity. In these equations, $X_i$ and $X_j$ represent the $i$th and $j$th samples in $X$. Distances were calculated using the pairwise_distances function from scikit-learn (version 1.1.2) (59).

$$\text{Jaccard distance}(X_i, X_j) = 1 - \frac{X_i \cap X_j}{X_i \cup X_j} \tag{1}$$

$$\text{Aitchison distance}(X_i, X_j) = \sqrt{\sum \left[ \text{CLR}(X_i) - \text{CLR}(X_j) \right]^2} \tag{2}$$

$$\text{Bray–Curtis dissimilarity}\,(X_i, X_j) = \frac{\sum |X_i - X_j|}{\sum (X_i + X_j)} \tag{3}$$

To calculate the Extra Trees dissimilarity, one simply determines the label of each terminal leaf into which a sample falls. This is recorded for each decision tree in the forest. These labels are then used to construct a binary matrix, $\overline{X}$, which simply describes the terminal leaves into which each sample falls. Each row of $\overline{X}$ is a sample, and each column is a leaf label. The similarity between two samples is then found by calculating how often two samples share a terminal node and dividing this count by the number of trees in the forest, $N$ (equation 4). Finally, the similarity is turned into a dissimilarity according to equation 5 (19, 20, 22, 41).

$$S(X_i, X_j) = \frac{\overline{X}_i \overline{X}_j^{\,T}}{N} \tag{4}$$

$$D(X_i, X_j) = \sqrt{1 - S(X_i, X_j)} \tag{5}$$

LANDMark returns this binary representation directly (25). To calculate the dissimilarity between samples in the LANDMark embedding, the Hamming distance can be used, since we are only interested in the positions where leaf labels are not shared. In these experiments, we used LANDMark version 1.2.2 and the implementation of the Extra Trees classifier found in scikit-learn (25, 59).

To use decision tree ensembles in an unsupervised manner, a second randomized data set is needed. To create the randomized data set, a copy of the original count matrix is created. Then, the counts of each ASV in the copy are randomly sampled without replacement (19, 60). The original data and randomized copy are then combined. To prevent Extra Trees and LANDMark from using information about the original class labels, the rows in the randomized data set are assigned a label of "Random," while original samples are assigned a label of "Original." The combined data set and the new labels are then used to train the decision tree ensemble. Dissimilarities are calculated by passing the original data to the model to determine which terminal leaves are associated with each sample.

**TreeOrdination: automating the identification of informative ASVs when using tree dissimilarities.** We have developed a workflow, TreeOrdination, which uses a relatively simple set of steps to create and analyze projections of high-dimensional amplicon sequencing data. This was done to investigate whether tree-based dissimilarities could be used to investigate beta diversity. The TreeOrdination and LANDMark source code and documentation are available at https://github.com/jrudar/LANDMark and https://github.com/jrudar/TreeOrdination.

TreeOrdination is a wrapper around the LANDMark and Uniform Manifold Approximation and Projection (UMAP) algorithms (25, 44). The default settings for LANDMark were used, except for the number of trees (which was set to 160), the use_nnet parameter (which was set to "False"), and the max_samples_tree parameter (which was set to 100) (25). Upon completion of training, the original training data are passed through the classifier, and the binary matrix describing which terminal leaves each sample falls into is extracted. Multiple LANDMark representations can be created and concatenated together. These representations can be used to train an Extra Trees classifier to predict the original class labels. In addition, the LANDMark representation is projected into a smaller 15-dimensional space using UMAP version 0.5.3 (44, 61). To create this projection, the min_dist parameter is set to 0.001, while the n_neighbors and metric parameters are set to 8 and Hamming, respectively. The PCA functionality of scikit-learn is then used to summarize the major directions of variation in UMAP representation (Fig. S1) (59). To understand which ASVs impact the locations of samples within the PCA projection, the original data are then used to train an Extra Trees regression model to predict the location of each sample in the projected space. This is necessary since calculating the impact that each ASV has on the location of each sample in the projected space using the original model is a computationally expensive task. Shapley scores are then used to calculate global feature importance scores and per-sample importance scores, respectively (46, 47). This method of investigating black-box models uses game theory to calculate the

importance and contribution of each feature to the final prediction. This is done by calculating how often a feature contributes to a prediction when included with a coalition of other features.

**Creation of positive and negative controls.** Positive and negative controls were created using a copy of the code found at https://github.com/cameronmartino/deicode-benchmarking/blob/master/simulations (27). This code was used in the simulations that tested robust centered log ratio transformation introduced by Martino et al. (27). Briefly, a synthetic data set containing 200 samples and 1,000 features was constructed according to the procedure found by Martino et al. (27). This was a positive control. A copy of this data set was made, and each feature column was independently permuted to create the negative control.

**Analysis of beta diversity and generalization performance in the case studies and synthetic data.** Samples were split into training and testing sets using repeated stratified 5-fold cross-validation. Five repeats were used. ASVs occurring in two or fewer training samples (in the case studies) and 5% of training samples (in the case of the control data sets) were identified. These ASVs were then removed from both the training and testing samples. This filtration step was taken since a reduction in the number of features can often lead to a more generalizable model (34, 62, 63). Following this, the size of each training library was calculated, and the 15th percentile of these sizes was found. Training and testing libraries were then rarefied so that each library was this size and those smaller than the 15th percentile were removed (7). The rarefied data were used for tests involving the presence-absence and proportion transformations. Tests using rCLR- and CLR-transformed data did not involve rarefaction, as these transformations can naturally handle compositions (6, 27, 64, 65). Count matrices were CLR- and rCLR-transformed using the scikit-bio (version 0.5.7) and DEICODE (version 0.2.4) packages, respectively (27, 66, 67). To produce the rCLR-transformed data set, 1,000 iterations of DEICODE's matrix completion algorithm were used. Following this, the $U$, $V$, and $S$ matrices (which are analogous to the sample loading, feature loading, and eigenvalue matrix of singular value decomposition) were extracted for further analysis (27). Randomized data for training unsupervised tree models were also created as described above, and the presence-absence, proportion, CLR, and rCLR transformations were applied.

The transformed training data were used as the input for a PERMANOVA (using 999 replications). The training data were also projected using principal coordinate analysis (PCoA)/robust principal component analysis (RPCA) and UMAP to create two-dimensional projections, which were used as input into another PERMANOVA analysis (using 999 replications). In this experiment, high $F$ statistics associated with significant $P$ values (defined as $P \leq 0.05$) provide evidence that differences exist between groups, and larger $F$ statistics provide evidence for a better separation between groups. PCoA projections were created using scikit-bio, while UMAP projections were created using the umap package (44, 66). In the case of rCLR-transformed data, this projection was automatically supplied in the form of the $U$ matrix. Unsupervised extremely randomized trees and unsupervised LANDMark projections were created by training an Extra Trees and LANDMark classifier. Both ensembles were constructed using 160 estimators. The use_nnet and max_samples_tree parameters of the LANDMark classifier were set to "False" and "100." Dissimilarities calculated using the unsupervised ensembles and the PCoA and UMAP projections of these dissimilarities were used as input into a PERMANOVA. Finally, TreeOrdination ordinations (using 160 estimators, 5 LANDMark classifiers, and 100 samples per tree) were created. These ordinations were also used as input into a PERMANOVA analysis.

Where appropriate, the transformed and projected training data were also used to train an Extra Trees and LANDMark classifier. The high-dimensional TreeOrdination embedding was used to train an Extra Trees classifier. The generalization performance of each model was measured using the balanced accuracy score (from scikit-learn) (59). This was chosen since it accounts for differences in class sizes when working with unbalanced data sets. To ascertain which models performed the best, a Wilcoxon signed rank test followed by the Benjamini-Hochberg correction, implemented in the Python statannotations package (version 0.5), was used (68).

To investigate which ASVs were different between sample types, we constructed a TreeOrdination model (as described above) using the Crohn's disease data. For this model, 70% of the samples were used for training data, while the remaining 30% were used for testing. Shapley scores for each group were calculated to determine which features impacted the location of samples in ordination space (28, 46, 47). The projection was visualized using the Python seaborn (version 0.11.2) and matplotlib (version 3.5.2) packages (69, 70).

**Analysis of generalization performance in the colorectal cancer data.** It was important to investigate how TreeOrdination and LANDMark behaved when faced with data where the differences between classes were not as clear. To carry out this investigation, we used a large colorectal cancer data set consisting of 490 samples (28). Since the goal of this experiment was to determine how well models that make multivariate cuts performed when the effect size between groups was smaller, a simplified analysis was conducted. The data were split into two groups: healthy versus colorectal cancer and healthy versus lesion (adenomas and colorectal cancers). The generalization performance using the ASV data and ASV data complemented using the fecal immunochemical test (FIT) was then measured (28). Five-fold stratified cross-validation with five repeats was used to split the samples from each group into training and testing data. ASVs found in fewer than 5% of samples in the training data were identified, and these ASVs were removed from both training and testing data. The counts for each sample were transformed into frequencies, with zero counts being replaced using the multiplicative replacement procedure, and the centered log ratio transformation was then applied (67). A random classifier (using a stratified sampling strategy), Extra Trees classifier (using 160 trees), random forest classifier (using 160 trees), LANDMark classifier (using 160 trees with each tree randomly selecting 100 samples and the use_nnet parameter set to False), and TreeOrdination (using 160 trees with each tree randomly selecting 100 samples) were used to measure generalization performance. The balanced accuracy and ROC-AUC scores were then calculated on each fold. The Wilcoxon signed rank test followed by the Benjamini-Hochberg correction for multiple comparisons was used to determine if the generalization performance between models was equivalent (68).

For each test (colorectal cancer versus healthy and lesion versus healthy), we measured the ability of LANDMark and TreeOrdination to identify colorectal cancers and lesions. This was done by first

determining the probability threshold that maximized the balanced accuracy score of the model concerning the prediction of lesions. To do this, we split each subset into training and testing folds using 5-fold stratified cross-validation with five repeats as described above. After training each model, the probability of colorectal cancer was determined using the test set. These probabilities were then averaged across folds. A list of probability thresholds (spaced evenly from 0 to 1.0 using increments of 0.01) was then used to determine the best possible cutoffs for the colorectal cancer versus healthy and healthy versus lesion models. Finally, we investigated which ASVs were different between sample types by constructing a TreeOrdination model (as described above). For this model, 80% of the samples were used for training data, while the remaining 20% were used for testing. Prediction probabilities were calculated using TreeOrdination. The subset of adenoma and colorectal cancer samples at or above this threshold were extracted, and healthy samples below this threshold in the test set were identified. These samples were then grouped into FIT-positive and FIT-negative samples and plotted. Shapley scores for each group were calculated to determine which features impacted the location of samples in ordination space (28, 46, 47).

**Data availability.** All data and code used in this analysis can be found at https://github.com/jrudar/Unsupervised-Decision-Trees. The raw sequences that were used in this study can be obtained from the Sequence Read Archive (accession numbers PRJNA450340 and SRP062005) (26, 28). The MetaWorks version 1.8.0 pipeline is available online at https://github.com/terrimporter/MetaWorks (56). The TreeOrdination and LANDMark source code and documentation are available at https://github.com/jrudar/LANDMark and https://github.com/jrudar/TreeOrdination.

## SUPPLEMENTAL MATERIAL

Supplemental material is available online only.

**SUPPLEMENTAL FILE 1**, PDF file, 0.4 MB.

## ACKNOWLEDGMENTS

We thank Katie McGee and Terri M. Porter for their thoughtful discussions during the development of LANDMark. We thank Marko Rudar for his thoughtful review of the manuscript. Finally, we thank our reviewers and editor for their time and comments. Their work has resulted in a considerably improved manuscript. J.R. is supported by funds from the Food from Thought project as part of Canada First Research Excellence Fund. M.H. received funding from the Government of Canada through Genome Canada and Ontario Genomics. G.B.G. is supported by a Natural Sciences and Engineering Research Council of Canada (NSERC) grant (RGPIN-2020-05733).

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
