## [Reviewer comments · Microbiology Spectrum]

Microbiology Spectrum

Decision Tree Ensembles Utilizing Multivariate Splits Are Effective at Investigating Beta-Diversity in Medically Relevant 16S Amplicon Sequencing Data

Josip Rudar, Geoffrey Golding, Stefan Kremer, and Mehrdad Hajibabaei

*Corresponding Authors: Josip Rudar and Mehrdad Hajibabaei,
University of Guelph*

Review Timeline:

Submission Date:	June 7, 2022
Editorial Decision:	June 29, 2022
Revision Received:	September 15, 2022
Editorial Decision:	October 7, 2022
Revision Received:	January 19, 2023
Accepted:	February 11, 2023

Editor: Jan Claesen

Reviewer(s): The reviewers have opted to remain anonymous.

Transaction Report:

DOI: <https://doi.org/10.1128/spectrum.02065-22>

June 29, 2022

Dr. Josip Rudar
University of Guelph
Integrative Biology
Guelph, Ontario
Canada

Re: Spectrum02065-22 (Decision Tree Ensembles Utilizing Multivariate Splits Are Effective at Investigating Beta-Diversity in Medically Relevant 16S Amplicon Sequencing Data)

Dear Dr. Josip Rudar:

Thanks for submitting your paper to Spectrum! Your manuscript has been evaluated by two independent Reviewers. Among the comments raised, the Reviewers pointed out that it is not entirely clear how the method described in the paper compares to existing methods and they provided suggestions to address this. I would be happy to consider a revised version of your manuscript in which the Reviewer comments are addressed in a point-by-point fashion. I realize this might involve running some additional data analyses, which could take longer than the standard allotted 60 days, and if you would like to request additional time for resubmission that would be agreeable.

Link Not Available

Sincerely,

Jan Claesen

Journals Department
Reviewer comments:

Reviewer #1 (Comments for the Author):

Rudar and colleagues describe the use of machine learning approaches to calculate and analyze beta-diversity metrics. There is considerable interest currently in using machine learning approaches with microbiome data. Unfortunately, I found the writing

and description of methods too challenging to follow. Furthermore, although the authors were re-analyzing previously published datasets, the new analysis is not at all aligned with the previous efforts, which made it difficult to compare the new results to the old.

1. The justification for the two datasets the authors analyzed was not strong (L103). The Flynn dataset is quite small with fewer than 20 subjects. As for the Forbes dataset, there are 5 conditions, but the number of subjects in each condition was between 19 and 23. I suspect there was significant overfitting of their models, yet there does not appear to be any cross validation or analysis to assess for overfitting or quality of the models. Also, a confounding issue for the Flynn dataset would be that the four sites are not independent of each other within each subject. I'm surprised that the authors didn't select a larger dataset such as the Baxter dataset that Schloss (corresponding author of the Flynn paper) has published numerous machine learning papers on that the authors could use for comparison (doi: 10.1128/mbio.03161-21, 10.1128/mBio.00434-20, 10.1186/s13073-016-0290-3, 10.1186/s40168-017-0366-3, 10.1186/s40168-016-0205-y)

2. One of the authors' motivations is to identify interactions between taxa using beta diversity metrics. But that isn't what they're designed to do. They're designed with measuring the dissimilarity between samples. Furthermore, I don't know how the authors think they can claim evidence for 'interactions' between features when all they have is a snapshot of subjects. There's no time series data here. Perhaps they're measuring co-occurrence or associations, but definitely not interactions.

3. One of my struggles is trying to figure out what the authors have found and how it's different from other methods. I would encourage the authors to use non CLR transformed data with "traditional" metrics like Bray-Curtis to see if these newer approaches gain anything over the simpler, traditional approaches. Furthermore, it would be nice to see a direct comparison - with the same ASVs - between what they are doing and what traditional methods have found. The authors acknowledge that data processing steps can have a significant impact on downstream steps and there's no consensus on things like ASVs or how to deal with compositional data. However, the authors have doubled down on using ASVs and using CLR rather than using other taxonomic levels that aren't so sparse or using traditional beta-diversity metrics like Bray-Curtis.

4. I found the manuscript was overloaded with Jargon and unsummarized data. For example on L107 the authors describe the four sites from Flynn that they used as "proximal lumen (RS) and mucosa (RB), the distal lumen (LS) and mucosa (LB)". The abbreviations have little connection to the site making it challenging for the reader to keep track of what is being compared. Furthermore, in the main paper there are 5 dense tables and 7 multipanel figures. I feel like the authors are giving the reader the data and asking the reader to draw conclusions. Finally, the machine learning methods descriptions is overly technical and I suspect the typical Microbiology Spectrum Reader will not be able to understand what the authors are trying to say.

5. It appears to me that the mark of the best method was providing the highest R2 value in a PERMANOVA test. Surely this could be spurious, right? A method can be overly optimistic and provide false positives. I suspect this might be the case if the models are overfit. The authors need a more objective way to assess the performance of their approaches.

6. The authors used a centered-log-ratio (CLR) transformation (L150). To do this a pseudocount needs to be added to values to avoid taking the logarithm of zero. What pseudocount was chosen and how did the choice of pseudocount affect the results?

7. I wonder whether the data were rarefied to a common number of sequences? Jaccard is highly susceptible to uneven sampling efforts. I suspect Aitchison is as well.

8. The description of the sequence curation is confusing at L137. It reads as though samples with confidence scores below 80% were removed from dataset. The Flynn dataset at least is a high quality dataset that likely has a very low sequencing error rate (full 2-fold coverage of the V4 region). I'm not sure why the authors would chose to remove such sequences. I would be curious what fraction of the sequences were removed.

Minor points...

L23. "16S rRNA data" / L103. "16S rRNA amplicon sequencing". Properly speaking the data analyzed were "16S rRNA gene" sequence data. No rRNA data are presented and the authors did not amplify rRNA - rather they sequenced the rRNA genes.

L151. "compositional (24,28). independent." something needs to be fixed here

Reviewer #2 (Comments for the Author):

Rudar et al present a comparison of methods to differentiate community level differences in the microbiome. They applied a tree-based algorithm (Landmark), and compared it to two supervised learning approaches, as well as Jaccard and Aitchison distance, two commonly used distance metrics. The authors also compared the ability to two ordination techniques to visually separate the data, applying PCoA and PCoA+UMAP to try and improve separation. They suggest that their tree-based algorithm

can best distinguish the biological groups of interest, and the dual ordination approach improves visual separation and encourage the use of the algorithm.

The authors provide a github link to two scripts written in base python to replicate their work.

Strengths

1. The authors are exploring novel techniques address a common frustration in microbiome research, where often important covariates only explain a small fraction of the overall variation in the microbial community. Similarly, the proposal of a secondary UMAP projection to amplify effects in PCoA space has the potential to enhance understanding
2. Comparisons between different machine learning techniques demonstrate the appropriateness of the LandMARK to separate the data
3. The use of real data in this simulation demonstrates this work is applicable at biologically relevant effect sizes and study sizes seen in the actual literature rather than small, toy datasets.

Places for Improvement

1. The article needs to clearly identify (1) what LandMARK is and (2) cite the previous paper. It was not entirely clear that this work built on a previously published algorithm nor what the abbreviation means when it appears
2. The article was submitted as a research article, but appears to present a new tool? algorithm? computational method? I did find the code, but its poorly documented and would be difficult to adapt into a new analysis by anyone by the authors. If this approach is intended for a wider audience (or even reproducibility beyond this single dataset), additional resources are needed. These might take the form of better documented code (explain the steps and the motivation), a wrapper function that takes an input table and framework and applies the appropriate transforms, a wrapper to perform the search space algorithm, and/or a tutorial...
I don't even know how to construct the environment based on the current available documentation.
3. I'm not sure you can really call the Landmark ordination unsupervised, since you're still using a supervised feature-selection. It seems like a circular process: you build this supervised classifier which selects features, convert it into an ordination and then claim it outperforms entirely unsupervised ordination...
If this is the case, maybe an additional comparison with PL-SDA might be appropriate.

If I'm misunderstanding, then I think the authors have missed more recent literature addressing some of their critiques of classic beta diversity metrics. For example, PCA on an ILR-transformed table (PhILR as an example) or DEICODE (a sparsity aware modified Aitchison) might be more appropriate comparisons.

4. I think the argument in the discussion that a single metric has to capture every aspect of the data is misleading. There may be a metric that best captures variation within a data set, but the claim of one true metric seems to dismiss a biological reality where different ecological mechanisms underpin perturbations in communities. This is not to say LANDmark-based ordination is not a useful tool, however, I think the discussion could and should be softened to allow room to test hypotheses around multiple potential ecological processes
5. I know this is a big ask, and may be beyond the scope of reasonable re-analysis, however, if the claim is that decision tree-based metrics better account for the correlated structure of the data, it seems simulated data with a known correlation structure would be a better proof of this than an unsubstantiated claim.

Other Suggestions

1. The mixing and introduction of ASV/OTU seems like a bit of a red herring. My recommendation would be to describe everything in the table as a "feature" since it could potentially be an ASV, OTU, species from metagenomic sequencing, or genome. The method should ultimately be agnostic to the type of feature, it just wants features
2. Please cite your tools. I shouldn't have to look at your figures to know this was a python implementation or go through your code to figure out the library defaults to know what was in your methods. This is especially important since the scipy Jaccard implementation is slightly different than Jaccard implementations in R.
Also because citation is a very cheap way to keep the resources we all rely on going, since citation shows a compelling need to funders for maintenance
3. Please check the text sizes on the final figures since most are difficult read at the current DPI. Also, if possible, please provide in-figure legends

Staff Comments:

Preparing Revision Guidelines

Please return the manuscript within 60 days; if you cannot complete the modification within this time period, please contact me. If you do not wish to modify the manuscript and prefer to submit it to another journal, please notify me of your decision immediately so that the manuscript may be formally withdrawn from consideration by Microbiology Spectrum.

Rudar et al present a comparison of methods to differentiate community level differences in the microbiome. They applied a tree-based algorithm (Landmark), and compared it to two supervised learning approaches, as well as Jaccard and Aitchison distance, two commonly used distance metrics. The authors also compared the ability to two ordination techniques to visually separate the data, applying PCoA and PCoA+UMAP to try and improve separation. They suggest that their tree-based algorithm can best distinguish the biological groups of interest, and the dual ordination approach improves visual separation and encourage the use of the algorithm. The authors provide a github link to two scripts written in base python to replicate their work.

Strengths

1. The authors are exploring novel techniques address a common frustration in microbiome research, where often important covariates only explain a small fraction of the overall variation in the microbial community. Similarly, the proposal of a secondary UMAP projection to amplify effects in PCoA space has the potential to enhance understanding
2. Comparisons between different machine learning techniques demonstrate the appropriateness of the LandMARK to separate the data
3. The use of real data in this simulation demonstrates this work is applicable at biologically relevant effect sizes and study sizes seen in the actual literature rather than small, toy datasets.

Places for Improvement

1. The article needs to clearly identify (1) what LandMARK is and (2) cite the previous paper. It was not entirely clear that this work built on a previously published algorithm nor what the abbreviation means when it appears
2. The article was submitted as a research article, but appears to present a new tool? algorithm? computational method?
I did find the code, but its poorly documented and would be difficult to adapt into a new analysis by anyone by the authors. If this approach is intended for a wider audience (or even reproducibility beyond this single dataset), additional resources are needed. These might take the form of better documented code (explain the steps and the motivation), a wrapper function that takes an input table and framework and applies the appropriate transforms, a wrapper to perform the search space algorithm, and/or a tutorial...
I don't even know how to construct the environment based on the current available documentation.
3. I'm not sure you can really call the Landmark ordination unsupervised, since you're still using a supervised feature-selection. It seems like a circular process:

you build this supervised classifier which selects features, convert it into an ordination and then claim it outperforms entirely unsupervised ordination... If this is the case, maybe an additional comparison with PL-SDA might be appropriate.

If I'm misunderstanding, then I think the authors have missed more recent literature addressing some of their critiques of classic beta diversity metrics. For example, PCA on an ILR-transformed table (PhILR as an example) or DEICODE (a sparsity aware modified Aitchison) might be more appropriate comparisons.

4. I think the argument in the discussion that a single metric has to capture every aspect of the data is misleading. There may be a metric that best captures variation within a data set, but the claim of one true metric seems to dismiss a biological reality where different ecological mechanisms underpin perturbations in communities. This is not to say LANDmark-based ordination is not a useful tool, however, I think the discussion could and should be softened to allow room to test hypotheses around multiple potential ecological processes
5. I know this is a big ask, and may be beyond the scope of reasonable re-analysis, however, if the claim is that decision tree-based metrics better account for the correlated structure of the data, it seems simulated data with a known correlation structure would be a better proof of this than an unsubstantiated claim.

Other Suggestions/

1. The mixing and introduction of ASV/OTU seems like a bit of a red herring. My recommendation would be to describe everything in the table as a "feature" since it could potentially be an ASV, OTU, species from metagenomic sequencing, or genome. The method should ultimately be agnostic to the type of feature, it just wants features
2. Please cite your tools. I shouldn't have to look at your figures to know this was a python implementation or go through your code to figure out the library defaults to know what was in your methods. This is especially important since the scipy Jaccard implementation is slightly different than Jaccard implementations in R. Also because citation is a very cheap way to keep the resources we all rely on going, since citation shows a compelling need to funders for maintenance
3. Please check the text sizes on the final figures since most are difficult read at the current DPI. Also, if possible, please provide in-figure legends

Response to Reviewers

- 1. The justification for the two datasets the authors analyzed was not strong (L103). The Flynn dataset is quite small with fewer than 20 subjects. As for the Forbes dataset, there are 5 conditions, but the number of subjects in each condition was between 19 and 23. I suspect there was significant overfitting of their models, yet there does not appear to be any cross validation or analysis to assess for overfitting or quality of the models. Also, a confounding issue for the Flynn dataset would be that the four sites are not independent of each other within each subject. I'm surprised that the authors didn't select a larger dataset such as the Baxter dataset that Schloss (corresponding author of the Flynn paper) has published numerous machine learning papers on that the authors could use for comparison (doi: 10.1128/mbio.03161-21, 10.1128/mBio.00434-20, 10.1186/s13073-016-0290-3, 10.1186/s40168-017-0366-3, 10.1186/s40168-016-0205-y)**

Thank you for this comment. In our initial manuscript, as you pointed out in your third and fourth comments, there was much to improve concerning clarity. The goal of this manuscript was to propose that decision tree ensemble classifiers (such as Extra Trees and LANDMark) could be used to investigate beta-diversity while concurrently identifying the ASVs (or OTUs, etc) that play a role in determining the location of samples in the projected space¹⁻⁴. To this end, we made large modifications to the manuscript. Furthermore, we focused our results on two case studies from these datasets to test how well these approaches can visualize and identify differences between the proximal lumen and mucosa of the human colon and between Crohn's Disease patients and healthy controls. 5-Fold Stratified Cross-Validation with 3 repeats was used to create different training and testing datasets. The balanced accuracy score was then calculated on the withheld data. In addition, we compared these results to those produced using a random classifier. Finally, while we would have liked to test an additional larger dataset, we did not have the time as we had to choose between this and the inclusion of positive and negative controls. We chose the latter since these can act as a rough proxy since a real dataset (the keyboard dataset) is used to create the positive controls. This simulation was created using the procedure and code from Martino et al. (2019)⁵.

We would also like to address your comment about overfitting in further detail. While some overfitting is likely to occur, we do not believe this is a problem in this experimental setup since we observed high balanced accuracy scores and these scores (Figure 3) were always more likely to be better than those observed when using a random classifier. Furthermore, randomization is a form of regularization when used with decision tree ensembles, which hopefully addresses your concern⁶. Finally, it is well known that individual decision trees tend to overfit the data and this often leads to poor generalization performance when these are used to predict class labels on unseen data points. However, if extensive randomization is used throughout the learning process to create a diverse assembly of individual trees, it becomes more likely that the classification errors made by each tree are uncorrelated^{3,7,8}. Averaging the results of all

the trees will then result in an improvement in generalization performance, even if using individual models which overfit.

2. One of the authors' motivations is to identify interactions between taxa using beta diversity metrics. But that isn't what they're designed to do. They're designed with measuring the dissimilarity between samples. Furthermore, I don't know how the authors think they can claim evidence for 'interactions' between features when all they have is a snapshot of subjects. There's no time series data here. Perhaps they're measuring co-occurrence or associations, but definitely not interactions.

Thank you for your comment. We believe there is a misunderstanding in what we are trying to accomplish in this analysis. You are correct in what we are measuring using tree-based dissimilarities: these are dependencies in the co-occurrence and possibly the abundance of ASVs. We have corrected the manuscript to reflect this and now explicitly state that tree-based dissimilarities can identify these dependencies during training. A significant number of changes have been made throughout the manuscript to improve readability and reader understanding of the terms used. Examples of these changes can be found on lines 21, 44, 93, 481-483.

3. One of my struggles is trying to figure out is what the authors have found and how it's different from other methods. I would encourage the authors to use non CLR transformed data with "traditional" metrics like Bray-Curtis to see if these newer approaches gain anything over the simpler, traditional approaches. Furthermore, it would be nice to see a direct comparison - with the same ASVs - between what they are doing and what traditional methods have found. The authors acknowledge that data processing steps can have a significant impact on downstream steps and there's no consensus on things like ASVs or how to deal with compositional data. However, the authors have doubled down on using ASVs and using CLR rather than using other taxonomic levels that aren't so sparse or using traditional beta-diversity metrics like Bray-Curtis.

Thank you for this comment. We chose to primarily focus on the two commonly used compositional approaches and the presence-absence transformation in this manuscript. The CLR (and now Robust CLR) transformations were used since these are increasingly used to address the problems associated with differences in read counts between taxa in each sample. While we agree that the use of more traditional metrics (like Bray-Curtis) has a place, during planning and testing we felt as though the inclusion of these metrics and other available transformations (CSS⁹, GMPR¹⁰, ILR¹¹) would take away from the main message of the manuscript: dissimilarities learned by tree-based ensembles can be used to investigate beta-diversity and identify differences in the composition of ecological and microbial communities. We did include this as a limitation in our concluding statement, however, while also providing a rationale for why and how traditional approaches can be used (speed, exploratory analysis). This change can be

found in lines 522 to 528.

4. I found the manuscript was overloaded with Jargon and unsummarized data. For example on L107 the authors describe the four sites from Flynn that they used as "proximal lumen (RS) and mucosa (RB), the distal lumen (LS) and mucosa (LB)". The abbreviations have little connection to the site making it challenging for the reader to keep track of what is being compared. Furthermore, in the main paper there are 5 dense tables and 7 multipanel figures. I feel like the authors are giving the reader the data and asking the reader to draw conclusions. Finally, the machine learning methods descriptions is overly technical and I suspect the typical Microbiology Spectrum Reader will not be able to understand what the authors are trying to say.

We agree with your comment and have significantly reduced the complexity of the manuscript. Since the main goal of this work was to investigate learned dissimilarities and present a proof-of-concept approach on how to use these to investigate beta-diversity, we focused our manuscript on two case studies: separating samples from the proximal lumen and mucosa and the Crohn's disease dataset. We also divided and re-wrote more complex sections of the methods in an attempt to reduce the amount of jargon. New figures and tables were generated which, hopefully, are better at conveying the message. Finally, we focused the manuscript on the main idea: a non-parametric approach to identify predictive ASVs and investigate community composition.

5. It appears to me that the mark of the best method was providing the highest R2 value in a PERMANOVA test. Surely this could be spurious, right? A method can be overly optimistic and provide false positives. I suspect this might be the case if the models are overfit. The authors need a more objective way to assess the performance of their approaches.

We have taken the approach used in Martino et al. (2019)⁵. The PerMANOVA¹² F-statistic was used to quantify the separation between groups for each transform and projection method in each case study (Figure 2). In addition, we tested if the configuration of samples between and within each of these projections could be similarly interpreted using the PROTEST procedure¹³ (Figure 1). We also tested if supervised models trained on the full and projected datasets could accurately predict the class label of withheld samples using 5-Fold Stratified Cross-Validation with 3 repeats (Figure 3 and 5). We hope that these changes to the manuscript will result in a more objective way to measure the performance of our approach.

6. The authors used a centered-log-ratio (CLR) transformation (L150). To do this a pseudocount needs to be added to values to avoid taking the logarithm of zero. What pseudocount was chosen and how did the choice of pseudocount affect the results?

Thank you for the comment. You are correct in that a pseudo-count was added. This was done using the multiplicative replacement procedure¹⁴. Since this is done automatically and is included in commonly used packages and algorithms^{5,15}, we did not feel it was necessary to address the effect of the pseudo-count at this time.

7. I wonder whether the data were rarefied to a common number of sequences? Jaccard is highly susceptible to uneven sampling efforts. I suspect Aitchison is as well.

Thank you for this comment. We did not rarefy the original data. Upon further consideration, we implemented this procedure and normalized each library such that the size was equal to the 15th percentile of all reads^{16,17}. This was only done for presence-absence data. Our methods now reflect these changes.

8. The description of the sequence curation is confusing at L137. It reads as though samples with confidence scores below 80% were removed from dataset. The Flynn dataset at least is a high quality dataset that likely has a very low sequencing error rate (full 2-fold coverage of the V4 region). I'm not sure why the authors would chose to remove such sequences. I would be curious what fraction of the sequences were removed.

Thank you for this comment. The ASVs were originally removed since the confidence in their assignment was below 80%. Originally, we only wanted to analyze ASVs which could be confidently assigned to a particular genus. However, upon further consideration, we re-ran the analysis with these ASVs included. Our methods section has been changed to reflect your concern.

Reviewer #2 (Comments for the Author):

Rudar et al present a comparison of methods to differentiate community level differences in the microbiome. They applied a tree-based algorithm (Landmark), and compared it to two supervised learning approaches, as well as Jaccard and Aitchison distance, two commonly used distance metrics. The authors also compared the ability to two ordination techniques to visually separate the data, applying PCoA and PCoA+UMAP to try and improve separation. They suggest that their tree-based algorithm can best distinguish the biological groups of interest, and the dual ordination approach improves visual separation and encourage the use of the algorithm.

The authors provide a github link to two scripts written in base python to replicate their work.

Strengths

1. The authors are exploring novel techniques address a common frustration in

microbiome research, where often important covariates only explain a small fraction of the overall variation in the microbial community. Similarly, the proposal of a secondary UMAP projection to amplify effects in PCoA space has the potential to enhance understanding

2. Comparisons between different machine learning techniques demonstrate the appropriateness of the LandMARK to separate the data

3. The use of real data in this simulation demonstrates this work is applicable at biologically relevant effect sizes and study sizes seen in the actual literature rather than small, toy datasets.

Thank you for the positive comments and suggestions. We hope that our improvements to the manuscript further clarify these strengths.

Places for Improvement

1. The article needs to clearly identify (1) what LandMARK is and (2) cite the previous paper. It was not entirely clear that this work built on a previously published algorithm nor what the abbreviation means when it appears

Thank you for this comment. We have modified our introduction to reflect this comment (see Lines 112 – 117)

2. The article was submitted as a research article, but appears to present a new tool? algorithm? computational method?

I did find the code, but its poorly documented and would be difficult to adapt into a new analysis by anyone by the authors. If this approach is intended for a wider audience (or even reproducibility beyond this single dataset), additional resources are needed. These might take the form of better documented code (explain the steps and the motivation), a wrapper function that takes an input table and framework and applies the appropriate transforms, a wrapper to perform the search space algorithm, and/or a tutorial...

I don't even know how to construct the environment based on the current available documentation.

Thank you for this comment. To address this concern, we have written a Python Notebook and provided better documentation on the code. In addition, we provided a wrapper, TreeOrdination, which will be available shortly at <https://github.com/jrudar/TreeOrdination>. A short tutorial on its use will be provided shortly after the submission of these revisions. Finally, we added which tools were used and their version numbers into the appropriate methods sections.

3. I'm not sure you can really call the Landmark ordination unsupervised, since you're still using a supervised feature-selection. It seems like a circular process: you build this supervised classifier which selects features, convert it into an ordination and then claim it outperforms entirely unsupervised ordination... If this is the case, maybe an additional comparison with PL-SDA might be appropriate.

Thank you for this comment. You are correct that LANDMark (and indeed most decision-tree ensembles) are supervised classifiers. However, they can be used in an “unsupervised” sense if one changes what the classifier is attempting to classify. In this manuscript, and in other research related to this procedure, the classifier does not make use of any of the original class labels and is only tasked to distinguish between real data and randomized data^{1,2,18,19}. The original data is then passed through the trained model to create an embedding or similarity matrix. To address your comment regarding the circular process, we created and tested a workflow that can be used to create the ordination in this manner. This workflow begins by learning N embeddings and then concatenating them. The concatenated vector is then used to train an Extra Trees classifier to predict the original class labels. In addition, a UMAP projection of the concatenated vector is created and the major axes of variation in this projection are summarized using PCA. We then use SAGE²⁰ and Shapley scores^{21,22} to identify the ASVs strongly impact the location of each sample in the projected space and quantify them. An example of these results can be found in Figures 4 – 7.

If I'm misunderstanding, then I think the authors have missed more recent literature addressing some of their critiques of classic beta diversity metrics. For example, PCA on an ILR-transformed table (PhILR as an example) or DEICODE (a sparsity aware modified Aitchison) might be more appropriate comparisons.

Thank you for your comment. We have included an additional analysis using sparsity-aware robust CLR transformation used by DEICODE. Unfortunately, we did not have time to include an analysis of the ILR transformation. However, we did not that exploring this approach is important in our concluding statement (Lines 526-528). The concluding statement also addresses a limitation in our study. Specifically, since this work represents a proof-of-concept we did not exhaustively test every transformation and projection method. However, we do believe that additional work is needed so that a better understanding of how machine learning can be used to investigate community composition using different transformations is needed.

4. I think the argument in the discussion that a single metric has to capture every aspect of the data is misleading. There may be a metric that best captures variation within a data set, but the claim of one true metric seems to dismiss a biological reality where different ecological mechanisms underpin perturbations in

communities. This is not to say LANDmark-based ordination is not a useful tool, however, I think the discussion could and should be softened to allow room to test hypotheses around multiple potential ecological processes

We added an additional discussion addressing that the choice in metric can have strengths and weaknesses depending on the dataset and what is being analyzed. We also suggested that this discussion is not settled and there is no consensus on which combination of approaches will be best since this is likely to be dataset-dependent (first paragraph of our discussion). In the concluding statement we also suggest that some metrics may be useful for exploratory analysis and, if projections account for a large fraction of the explained variance, there may be little value that is added using learned dissimilarities due to their additional complexity. We hope that these additional discussion points satisfy your concerns and we thank you for the comment.

5. I know this is a big ask, and may be beyond the scope of reasonable re-analysis, however, if the claim is that decision tree-based metrics better account for the correlated structure of the data, it seems simulated data with a known correlation structure would be a better proof of this than an unsubstantiated claim.

We included a positive and negative control using the procedure described in Martino et al. (2019)⁵.

Other Suggestions

1. The mixing and introduction of ASV/OTU seems like a bit of a red herring. My recommendation would be to describe everything in the table as a "feature" since it could potentially be an ASV, OTU, species from metagenomic sequencing, or genome. The method should ultimately be agnostic to the type of feature, it just wants features

We made major modifications in the language throughout the manuscript to better reflect that ASVs are features. We hope that these changes improve the clarity of the manuscript and convey to others that this approach can be agnostic to the type of features.

2. Please cite your tools. I shouldn't have to look at your figures to know this was a python implementation or go through your code to figure out the library defaults to know what was in your methods. This is especially important since the scipy Jaccard implementation is slightly different than Jaccard implementations in R. Also because citation is a very cheap way to keep the resources we all rely on going, since citation shows a compelling need to funders for maintenance

Thank you for this comment. We re-wrote the methods section and did a better job at describing which packages were used and the versions of the packages used.

3. Please check the text sizes on the final figures since most are difficult read at the current DPI. Also, if possible, please provide in-figure legends

New figures were generated. We hope that these are clearer and more informative than previously used ones.

References

1. Alhusain, L. & Hafez, A. M. Cluster ensemble based on Random Forests for genetic data. *BioData Mining* **10**, 37 (2017).
2. Rhodes, J. S., Cutler, A. & Moon, K. R. Geometry- and Accuracy-Preserving Random Forest Proximities. (2022) doi:10.48550/ARXIV.2201.12682.
3. Rudar, J., Porter, T. M., Wright, M., Golding, G. B. & Hajibabaei, M. LANDMark: An ensemble approach to the supervised selection of biomarkers in high-throughput sequencing data. *BMC Bioinformatics* **23**, 110 (2022).
4. Breiman, L. Random Forests. *Machine Learning* **45**, 5–32 (2001).
5. Martino, C. *et al.* A Novel Sparse Compositional Technique Reveals Microbial Perturbations. *mSystems* **4**, (2019).
6. Mentch, L. & Zhou, S. Randomization as Regularization: A Degrees of Freedom Explanation for Random Forest Success. *Journal of Machine Learning Research* **21**, 171:1-171:36 (2020).
7. Kuncheva, L. A Bound on Kappa-Error Diagrams for Analysis of Classifier Ensembles. *IEEE Transactions on Knowledge and Data Engineering* **25**, 494–501 (2013).
8. Kuncheva, L. I. & Rodriguez, J. J. Classifier ensembles with a random linear oracle. *IEEE Transactions on Knowledge and Data Engineering* **19**, 500–508 (2007).
9. Paulson, J. N., Stine, O. C., Bravo, H. C. & M, P. Robust methods for differential abundance analysis in marker gene surveys. *Nature Methods* **10**, 1200–1202 (2013).

10. Chen, L. *et al.* GMPR: A robust normalization method for zero-inflated count data with application to microbiome sequencing data. *PeerJ* **6**, e4600 (2018).
11. Silverman, J. D., Washburne, A. D., S, M. & David, L. A. A phylogenetic transform enhances analysis of compositional microbiota data. *eLife* **6**, 21887 (2017).
12. Anderson, M. J. & Walsh, D. C. I. PERMANOVA, ANOSIM, and the Mantel test in the face of heterogeneous dispersions: What null hypothesis are you testing? *Ecological Monographs* **83**, 557–574 (2013).
13. Jackson, D. A. PROTEST: A PROcrustean Randomization TEST of community environment concordance. *Écoscience* **2**, 297–303 (1995).
14. Martín-Fernández, J. A., Barceló-Vidal, C. & Pawlowsky-Glahn, V. Dealing with Zeros and Missing Values in Compositional Data Sets Using Nonparametric Imputation. *Mathematical Geology* **35**, 253–278 (2003).
15. team, T. scikit-bio development. scikit-bio: A Bioinformatics Library for Data Scientists, Students, and Developers. (2022).
16. McMurdie, P. J. & Holmes, S. Waste Not, Want Not: Why Rarefying Microbiome Data Is Inadmissible. *PLoS Computational Biology* **10**, 1003531 (2014).
17. Weiss, S. *et al.* Normalization and microbial differential abundance strategies depend upon data characteristics. *Microbiome* **5**, (2017).
18. Dalleau, K., Couceiro, M. & Smail-Tabbone, M. Unsupervised Extremely Randomized Trees. in *Advances in Knowledge Discovery and Data Mining* (eds. Phung, D. *et al.*) 478–489 (Springer International Publishing, 2018).

19. Bernard, S., Cao, H., Sabourin, R. & Heutte, L. Random Forest for Dissimilarity-based Multi-view Learning. in *Handbook of Pattern Recognition and Computer Vision* 119–138 (WORLD SCIENTIFIC, 2020). doi:10.1142/9789811211072_0007.
20. Covert, I. C., Lundberg, S. & Lee, S.-I. Understanding Global Feature Contributions with Additive Importance Measures. in *Proceedings of the 34th International Conference on Neural Information Processing Systems* (Curran Associates Inc., 2020).
21. Lundberg, S. M. *et al.* From local explanations to global understanding with explainable AI for trees. *Nat Mach Intell* **2**, 56–67 (2020).
22. Lundberg, S. M. & Lee, S. A Unified Approach to Interpreting Model Predictions. in *31st Conference on Neural Information Processing Systems (NIPS 2017)* (2017).

October 7, 2022

Dr. Josip Rudar
University of Guelph
Integrative Biology
Guelph, Ontario
Canada

Re: Spectrum02065-22R1 (Decision Tree Ensembles Utilizing Multivariate Splits Are Effective at Investigating Beta-Diversity in Medically Relevant 16S Amplicon Sequencing Data)

Dear Dr. Josip Rudar:

I sent your manuscript back to the original Reviewers for evaluation and I agree with them that their comments from the first submission have not been fully addressed. Could you please address all comments from both the previous and current evaluation and provide a marked-up manuscript that allows the Reviewers and me to follow the edits made. I'm sorry, but unless these problems are fully addressed, we will have to reject your paper.

Link Not Available

Sincerely,

Jan Claesen

Journals Department
Reviewer comments:

Reviewer #1 (Comments for the Author):

Review

I appreciate the authors' attempts to respond to my comments and those of the other reviewer. On the whole, I do not feel that the authors were fully responsive to our comments.

This is exemplified by the fact that they did not submit a marked up version of the manuscript. Rather, their marked up manuscript is a bunch of comments point to where the reviewers made comments. It is not possible to look at the marked up version of the manuscript and see what has changed.

Furthermore, in their response to the reviews, the authors mentioned several times that there wasn't enough time to complete an analysis. Science is not a timed trial. Work needs to be done to couch the analysis in the proper context. I am still unclear why the authors chose the relatively small Flynn dataset with large effect size rather than a larger, more realistic dataset with a more modest effect size that has already been studied by machine learning approaches (e.g. the Baxter dataset I mentioned). Also, just because two reviewers and an editor signed off on an earlier manuscript does not exempt it from further scrutiny. The effect of adding a pseudocount is potentially problematic and could inflate or deflate the ability to detect real differences.

Rather than brushing off my comments, the authors need to test for overfitting the authors need to split the data to establish a training and testing set and then repeat the split a large number of times (e.g. 100 times) to assess for performance and overfitting between the training and testing datasets.

The manuscript is jargon laden and will be impossible for most microbiologists to make sense of. If the authors want microbiologists to use this work in their own, then the authors need to make it much more accessible.

The data are still poorly summarized and presented in the 3 large tables and 7 large figures.

Reviewer #2 (Comments for the Author):

Thank you to the authors for their substantial revisions and clarifications around versions. However, I have some remaining concerns.

In my original revision in point 2, I raised the issue of a lack of reproducibility and the TreeOrdination github repo was offered as a solution. Since this paper is supposed to sway the reader to use the new method, it seemed critical to know it actually worked and if I could apply the method using one of my favorite datasets.

I was unable to install the Landmark package in a clean python 3.10 conda environment. (I got an error message that scikit-learn version 1.1.4 was required, and the closest available version was 1.1.2.) I cannot reasonably review a method that I can't run. The notebooks that were provided with this manuscript are quite dense and poorly documented, even for people familiar with python. The tutorial provides very little information about what format the data needs to be in or any pre-processing steps required. I anticipate that even if I could get the package installed, it would be a challenge to coerce the example data on my computer into a format compatible with the tool provided.

I would highly recommend inviting a student, bioinformatician, or biostatistician to collaborate and provide a review for the documentation and tool. They should be able to install the libraries, run the tutorial code, and interpret the results without needing to ask the original team questions. You might also discuss with them how likely they would be to use your current wrapper version and how it would integrate with their current workflows.

I also don't feel you've satisfied issues around jargon that the other reviewer and I raised. "Hot embedding" (lines 229-231) is unlikely to be familiar to readers of this journal, could you clarify this term? The SAGE/Shiplely methods used for identifying key features is also not well described. Since these were first published within the last 5 years, it's possible that even experts in feature selection may not be familiar with these methods.

In addition, given the move to DEICODE and the focus on the rPCA as a point of comparison, it seems like a natural comparison should have been feature loadings in that space.

In the response to reviewers document, it was mentioned that this is a "proof of concept", and therefore did not represent an exhaustive test. Based on that, the concluding statement about other metrics being more appropriate still seems very bold.

Additional remaining issues:

- The scikit libraries are canonically lowercased. (scikit-learn and scikit-bio).
- The text in several figures is still too small to read at scale in what will probably be the displayed figure. This is a problem in figure S1, figure 5, and several others.
- You might consider long-transforming p-values when displaying them in boxplots, or possibly presenting them as a swarm plot.
- Permutative p-values should always be reported with the number of permutations

Staff Comments:

Preparing Revision Guidelines

Please return the manuscript within 60 days; if you cannot complete the modification within this time period, please contact me. If you do not wish to modify the manuscript and prefer to submit it to another journal, please notify me of your decision immediately so that the manuscript may be formally withdrawn from consideration by Microbiology Spectrum.

Response to Reviewers

- 1. The justification for the two datasets the authors analyzed was not strong (L103). The Flynn dataset is quite small with fewer than 20 subjects. As for the Forbes dataset, there are 5 conditions, but the number of subjects in each condition was between 19 and 23. I suspect there was significant overfitting of their models, yet there does not appear to be any cross validation or analysis to assess for overfitting or quality of the models. Also, a confounding issue for the Flynn dataset would be that the four sites are not independent of each other within each subject. I'm surprised that the authors didn't select a larger dataset such as the Baxter dataset that Schloss (corresponding author of the Flynn paper) has published numerous machine learning papers on that the authors could use for comparison (doi: 10.1128/mbio.03161-21, 10.1128/mBio.00434-20, 10.1186/s13073-016-0290-3, 10.1186/s40168-017-0366-3, 10.1186/s40168-016-0205-y)**

Thank you for this comment. In response to your comment we removed our analysis of the Flynn dataset and included an analysis of the Baxter dataset. We also highlighted why the analysis of these datasets is important (Lines 267-290). Before addressing this comment more directly, we would like to convey that a detailed analysis of these datasets is outside the scope of this study since the aim of our work is to show that metric learning can be used to generate better ordinations. We also intended to demonstrate that combining metric learning with modern model introspection algorithms can unify the analysis of community composition by identifying features (such as ASVs, OTUs, etc) which are associated with the overall variation in ordination and features which impact the placement of individual samples in the ordination. Examples of such analyses can be found in Figures 11 to 14. We also agree that overfitting is likely to be a problem in most machine learning analysis and that proper cross-validation is necessary to ensure that predictive models generalize well. To this end we clearly stated where cross-validation was added to our analysis (Lines 575-577, 592-593, 611-613, and Figures 1, 2, and 3 as examples of edits we made to improve clarity about how data was generated). To further elaborate on our use of cross-validation, we used five-fold stratified cross-validation with five repeats to test our models since we felt that 100 folds on these datasets would be excessive given their size. Although we chose not to use 100 folds we did address additional potential issues that could lead to overly optimistic models. For example, we took extra caution to ensure that any transformation applied to the training set would also be applied to the test set (examples can be found on Lines 510-512, 577-578). Furthermore, we did not test generalization performance using PCoA and RPCA transformed data since this would require us to project the entire dataset before splitting it into training and testing data. As we have discussed in the manuscript (Lines 761-763, 1312-1320) these methods incorporate information about the test data when optimizing the loss and therefore this should be avoided to prevent overly optimistic results. Finally, we included tests using a random classifier to ensure that the generalization performance of models in the Baxter dataset (measured using balanced accuracy) was always better than this baseline (Figure 6 and 8).

Finally, we would like to clarify how overfitting is addressed in TreeOrdination, LANDMark, and Random Forests/Extremely Randomized Trees. Recent work has demonstrated that randomization in the selection of features can be considered a form of regularization when used with decision tree ensembles¹. In these models, features which are not selected at nodes are effectively zeroed-out. In LANDMark, this is taken a step further and linear models are L1 and L2 regularized while neural network models (although not used here) are regularized using

dropout. Nodes in LANDMark are usually trained on a bootstrapped selection of samples and are therefore trained using missing information². Due to the extensive randomization that is used throughout the learning process, a diverse assembly of individual trees is created and it becomes more likely that the classification errors made by each tree are uncorrelated²⁻⁴. For these reasons we are confident that LANDMark and TreeOrdination models do not severely overfit and this is supported by our results (See Figures 2, 3, 5, 6, and 7).

2. One of the authors' motivations is to identify interactions between taxa using beta diversity metrics. But that isn't what they're designed to do. They're designed with measuring the dissimilarity between samples. Furthermore, I don't know how the authors think they can claim evidence for 'interactions' between features when all they have is a snapshot of subjects. There's no time series data here. Perhaps they're measuring co-occurrence or associations, but definitely not interactions.

Thank you for your comment. We believe there is a misunderstanding in what we are trying to accomplish in this analysis. When using tree-based dissimilarities we are potentially modelling dependencies in the co-occurrence and possibly the abundance of ASVs (or any other set of features used to train the model). We have corrected the manuscript to reflect this and now explicitly state that tree-based dissimilarities can potentially identify these dependencies during training. Examples of these changes can be found on Lines 111-113, 119-121, 1110-1112, 1176-1178.

3. One of my struggles is trying to figure out is what the authors have found and how it's different from other methods. I would encourage the authors to use non CLR transformed data with "traditional" metrics like Bray-Curtis to see if these newer approaches gain anything over the simpler, traditional approaches. Furthermore, it would be nice to see a direct comparison - with the same ASVs - between what they are doing and what traditional methods have found. The authors acknowledge that data processing steps can have a significant impact on downstream steps and there's no consensus on things like ASVs or how to deal with compositional data. However, the authors have doubled down on using ASVs and using CLR rather than using other taxonomic levels that aren't so sparse or using traditional beta-diversity metrics like Bray-Curtis.

Thank you for this comment. In our latest revision we have included the Bray-Curtis measure and an analysis using proportions (Lines 329-338, Figures 1-5). We also addressed important limitations of the Robust CLR and PCoA ordinations. Using these can result in data leakage (Lines 751-755, 1142-1150). To address this issue we avoided using ordinations created using PCoA and RPCA to train machine learning models and instead used data transformed into proportions as input into UMAP with the metric parameter set to "braycurtis". This did result in informative ordinations in the Crohn's Disease case study but the separation between groups in these ordinations (measured using the F-statistic) was smaller than with CLR and presence-absence transformed data (Figure 4). We also added lines 1196-1199, 1456-1460, 1470-1476 to address potential limitations of this work and future directions. We hope that this additional analysis helps alleviate the reviewers concern.

Finally, we would like to address your comment over the novelty of this work. We want to emphasize that the approaches described here (LANDMark and TreeOrdination) are general

purpose algorithms that can be applied to any dataset. Using ASVs and the transformations here just serve as a way to provide consistent data to train the algorithms. So while we agree with your comment that there is no consensus on the use of ASVs or how to deal with compositional data, adding experiments such as these would take away from the main message of this work: 1) metric learning can be used to investigate community differences in complex ecological datasets and 2) metric learning and model introspection methods can be used to effectively identify which features are associated with variation in ordinations at both a global and local scale. This allows a single model, such as TreeOrdination, to be used to not only model beta-diversity but also understand how differences in the abundance or presence of organisms influences the position of samples in these spaces (Figure 10-13 are good examples of this). Finally, unlike PCoA and UMAP, a dissimilarity which is learned is less likely to be impacted by the assumptions of commonly used dissimilarity and distance measures⁵⁻⁷.

4. I found the manuscript was overloaded with Jargon and unsummarized data. For example on L107 the authors describe the four sites from Flynn that they used as "proximal lumen (RS) and mucosa (RB), the distal lumen (LS) and mucosa (LB)". The abbreviations have little connection to the site making it challenging for the reader to keep track of what is being compared. Furthermore, in the main paper there are 5 dense tables and 7 multipanel figures. I feel like the authors are giving the reader the data and asking the reader to draw conclusions. Finally, the machine learning methods descriptions is overly technical and I suspect the typical Microbiology Spectrum Reader will not be able to understand what the authors are trying to say.

We agree with your comment and have significantly reduced the complexity of the manuscript. We have extensively re-written large portions of the manuscript to reduce the amount of field-specific jargon. We have also removed comparisons with other classifiers so that the analysis is as straightforward as possible to the reader. We have removed many of the dense tables and multipanel figures and we hope that the new figures and their captions capture the main idea of each figure and study more clearly.

5. It appears to me that the mark of the best method was providing the highest R2 value in a PERMANOVA test. Surely this could be spurious, right? A method can be overly optimistic and provide false positives. I suspect this might be the case if the models are overfit. The authors need a more objective way to assess the performance of their approaches.

We have taken the approach used in Martino et al. (2019)⁸. The PerMANOVA⁹ F-statistic was used to quantify the separation between groups for each transform and projection method in each case study (Figure 1 and 4). In addition, we provide the distribution of p-values for the synthetic tests and Crohn's Disease case study: Lines 611-613 and Figure 4. We also provide cross-validated balanced accuracy scores for these tests (Figure 2, 3, and 5) and for the Baxter dataset (Figure 6 and 8).

6. The authors used a centered-log-ratio (CLR) transformation (L150). To do this a pseudocount needs to be added to values to avoid taking the logarithm of zero. What pseudocount was chosen and how did the choice of pseudocount affect the results?

Thank you for the comment. You are correct in that a pseudo-count was added. This was done using the multiplicative replacement procedure¹⁰. We did not feel it was appropriate to address the effect of the pseudo-count at this time since these tests would be outside the scope of this work. The main aim of this study was to determine if metric learning can be successfully applied to ecological datasets and our results support this position. However, we do agree that this is an important consideration and should be addressed in future work that is specifically focused on this issue.

7. I wonder whether the data were rarefied to a common number of sequences? Jaccard is highly susceptible to uneven sampling efforts. I suspect Aitchison is as well.

Thank you for this comment. We did not rarefy the original data. Upon further consideration, we implemented this procedure and normalized each library such that the size was equal to the 15th percentile of all reads^{11,12}. The library size to which samples would be normalized was calculated using the training dataset and this value applied to both the training and test datasets (libraries smaller than this were removed from the analysis). This was only done for presence-absence data and data which was to be used to measure the Bray-Curtis dissimilarity between samples. Our methods now reflect these changes (Lines 514-523).

8. The description of the sequence curation is confusing at L137. It reads as though samples with confidence scores below 80% were removed from dataset. The Flynn dataset at least is a high quality dataset that likely has a very low sequencing error rate (full 2-fold coverage of the V4 region). I'm not sure why the authors would chose to remove such sequences. I would be curious what fraction of the sequences were removed.

Thank you for this comment. The ASVs were originally removed since the confidence in their assignment was below 80%. Originally, we only wanted to analyze ASVs which could be confidently assigned to a particular genus. However, upon further consideration, we re-ran the analysis with all ASVs included. If an ASV could not be confidently identified, it was set as "Unclassified". Our methods section has been changed to reflect this concern (Lines 303-304).

9. I appreciate the authors' attempts to respond to my comments and those of the other reviewer. On the whole, I do not feel that the authors were fully responsive to our comments.

We hope that our modifications to your initial comments are more insightful and address this concern more fully. We appreciate the time and effort taken to review this manuscript and hope that you see this as a substantial improvement over the original work.

10. This is exemplified by the fact that they did not submit a marked up version of the manuscript. Rather, their marked up manuscript is a bunch of comments point to where the reviewers made comments. It is not possible to look at the marked up version of the manuscript and see what has changed.

We apologize for this oversight and have now provided a marked up manuscript with extensive revisions for your review.

11. Furthermore, in their response to the reviews, the authors mentioned several times that there wasn't enough time to complete an analysis. Science is not a timed trial. Work needs to be done to couch the analysis in the proper context. I am still unclear why the authors chose the relatively small Flynn dataset with large effect size rather than a larger, more realistic dataset with a more modest effect size that has already been studied by machine learning approaches (e.g. the Baxter dataset I mentioned). Also, just because two reviewers and an editor signed off on an earlier manuscript does not exempt it from further scrutiny. The effect of adding a pseudocount is potentially problematic and could inflate or deflate the ability to detect real differences.

This dataset was originally chosen as a case study which could be used to validate our approach. However, we see the logic in your argument and have therefore removed the small Flynn dataset and used, as per your recommendation, the Baxter dataset. With respect to the pseudocount problem, we agree that this work should be done but it is outside the scope of this work for the aforementioned reasons. In addition, our overall results are consistent with that of the wider literature regarding IMIDs and colorectal cancer with many of the ASVs detected as different between groups by TreeOrdination having known links to these conditions.

12. Rather than brushing off my comments, the authors need to test for overfitting the authors need to split the data to establish a training and testing set and then repeat the split a large number of times (e.g. 100 times) to assess for performance and overfitting between the training and testing datasets.

Given the size of these datasets a large number of cross-validation folds is unlikely to be necessary. Furthermore, there is no consensus as to how many folds are necessary. For example, in a recently published work on the gut-microbiome utilized 5-fold, 8-fold, 10-fold, and 100-fold cross-validation¹³⁻¹⁸. We did address this concern by expanding the number of repeats (five) and using five stratified cross-validation. We have also taken additional measures to limit possible overfitting. We also addressed how the metric learning approaches taken here address the problem of overfitting through regularization and randomization (see our response to Comment 1).

13. The manuscript is jargon laden and will be impossible for most microbiologists to make sense of. If the authors want microbiologists to use this work in their own, then the authors need to make it much more accessible.

The data are still poorly summarized and presented in the 3 large tables and 7 large figures.

We have made extensive modifications to the manuscript in an attempt to simplify the jargon, properly summarize the results in the figures, and clarify the goals of this work. We apologize that this was not addressed fully in the first round of revisions.

Reviewer #2 (Comments for the Author):

Rudar et al present a comparison of methods to differentiate community level differences in the microbiome. They applied a tree-based algorithm (Landmark), and compared it to two supervised learning approaches, as well as Jaccard and Aitchison distance, two commonly

used distance metrics. The authors also compared the ability to two ordination techniques to visually separate the data, applying PCoA and PCoA+UMAP to try and improve separation. They suggest that their tree-based algorithm can best distinguish the biological groups of interest, and the dual ordination approach improves visual separation and encourage the use of the algorithm.

The authors provide a github link to two scripts written in base python to replicate their work.

Strengths

1. The authors are exploring novel techniques address a common frustration in microbiome research, where often important covariates only explain a small fraction of the overall variation in the microbial community. Similarly, the proposal of a secondary UMAP projection to amplify effects in PCoA space has the potential to enhance understanding

2. Comparisons between different machine learning techniques demonstrate the appropriateness of the LandMARK to separate the data

3. The use of real data in this simulation demonstrates this work is applicable at biologically relevant effect sizes and study sizes seen in the actual literature rather than small, toy datasets.

Thank you for the positive comments and suggestions. We hope that our improvements to the manuscript work to further clarify these strengths.

Places for Improvement

1. The article needs to clearly identify (1) what LandMARK is and (2) cite the previous paper. It was not entirely clear that this work built on a previously published algorithm nor what the abbreviation means when it appears

Thank you for this comment. We have modified our introduction to reflect this comment (see Lines 250 – 259)

2. The article was submitted as a research article, but appears to present a new tool? algorithm? computational method?

I did find the code, but its poorly documented and would be difficult to adapt into a new analysis by anyone by the authors. If this approach is intended for a wider audience (or even reproducibility beyond this single dataset), additional resources are needed. These might take the form of better documented code (explain the steps and the motivation), a wrapper function that takes an input table and framework and applies the appropriate transforms, a wrapper to perform the search space algorithm, and/or a tutorial...

I don't even know how to construct the environment based on the current available documentation.

In my original revision in point 2, I raised the issue of a lack of reproducibility and the TreeOrdination github repo was offered as a solution. Since this paper is supposed to sway the reader to use the new method, it seemed critical to know it actually worked and if I

could apply the method using one of my favorite datasets.

I was unable to install the Landmark package in a clean python 3.10 conda environment. (I got an error message that scikit-learn version 1.1.4 was required, and the closest available version was 1.1.2.) I cannot reasonably review a method that I can't run.

The notebooks that were provided with this manuscript are quite dense and poorly documented, even for people familiar with python. The tutorial provides very little information about what format the data needs to be in or any pre-processing steps required. I anticipate that even if I could get the package installed, it would be a challenge to coerce the example data on my computer into a format compatible with the tool provided.

Thank you for this comment. To address this concern, we have written Python scripts that separate the data generation and analysis portions of the analysis. We have also better documented the code in the form of comments. We apologize for you not being able to get TreeOrdination and LANDMark working correctly. We have updated these packages to reflect this problem and hope that the problem is solved. Unfortunately, some of the dependencies (such as scikit-bio) needed to be modified to make this work. We will provide an updated set of instructions on how to set up an environment shortly after the submission of this revision. We will also improve our documentation on how to use TreeOrdination shortly after the submission of this revision. Finally, we have added which tools were used and their version numbers into the appropriate methods sections.

3. I'm not sure you can really call the Landmark ordination unsupervised, since you're still using a supervised feature-selection. It seems like a circular process: you build this supervised classifier which selects features, convert it into an ordination and then claim it outperforms entirely unsupervised ordination...

If this is the case, maybe an additional comparison with PL-SDA might be appropriate.

Thank you for this comment. You are correct that LANDMark (and indeed most decision-tree ensembles) are supervised classifiers. However, they can be used in an “unsupervised” sense if one changes what the classifier is attempting to classify. In this manuscript, and in other research related to this procedure, the classifier does not make use of any of the original class labels and is only tasked to distinguish between real data and randomized data^{7,19–21}. The original data is then passed through the trained model to create a high-dimensional embedding or similarity matrix. To be clear on how this process works we revised our methods section to describe how decision tree ensembles can be used in an unsupervised manner (Lines 339-495).

If I'm misunderstanding, then I think the authors have missed more recent literature addressing some of their critiques of classic beta diversity metrics. For example, PCA on an ILR-transformed table (PhILR as an example) or DEICODE (a sparsity aware modified Aitchison) might be more appropriate comparisons.

In addition, given the move to DEICODE and the focus on the rPCA as a point of comparison, it seems like a natural comparison should have been feature loadings in that space.

In the response to reviewers document, it was mentioned that this is a "proof of concept", and therefore did not represent an exhaustive test. Based on that, the concluding statement about other metrics being more appropriate still seems very bold.

Thank you for your comments. We have included an additional analysis using the sparsity-aware robust CLR transformation used by DEICODE. We did not include an analysis of the ILR transformation as we limited our investigation to more commonly used approaches. However, this transformation should be included in a more detailed and wider-ranging comparison. While using the feature loading matrix from RPCA seems like a natural point of comparison, there is an issue with this algorithm that needs to be considered. RPCA is unable to project new and unseen data into the lower dimensional space specified by the investigator. Therefore, there is no reliable way to assess generalization performance of this method (and PCoA) since these methods cannot be used to create independent training and testing data without causing data leakage. Due to this, the information in the feature loadings is difficult to assess. For this reason, we felt it wise not to conduct this comparison since it could result in misleading and overly optimistic results. Instead, we focused on how TreeOrdination and Shapley scores could be used identify important features and compared these results with what is reported in the literature (Figures 10-13 and lines 1214-1464). However, that does not mean that RPCA is not valuable. Since RPCA can identify uninformative features efficiently we do think it would be interesting to do a follow-up study to examine how RPCA can work for feature selection. Specifically, it might be possible to use RPCA as a first-pass filter before a more computationally intensive feature selection algorithm is applied. Finally, to address your final comment, we hope to have tapered our enthusiasm for this approach and focused our discussion on how LANDMark and TreeOrdination address an important gap when it comes to the analysis of community composition.

4. I think the argument in the discussion that a single metric has to capture every aspect of the data is misleading. There may be a metric that best captures variation within a data set, but the claim of one true metric seems to dismiss a biological reality where different ecological mechanisms underpin perturbations in communities. This is not to say LANDmark-based ordination is not a useful tool, however, I think the discussion could and should be softened to allow room to test hypotheses around multiple potential ecological processes

We added an additional discussion addressing a limitation of our work (Lines 1196-1199, 1456-1460, 1470-1476). We also amended our discussion to state that the choice of transformation is likely to be dataset dependent and it is important for the investigator to determine which approach is best given the data and the question being asked (Lines 1196-1199).

5. I know this is a big ask, and may be beyond the scope of reasonable re-analysis, however, if the claim is that decision tree-based metrics better account for the correlated structure of the data, it seems simulated data with a known correlation structure would be a better proof of this than an unsubstantiated claim.

We included a positive and negative control using the procedure described in Martino et al. (2019)⁸. The methods used for these tests are outlined in lines 499-506/507-549 and the results reported in on lines 610-763 Figures 1 – 3, Table 1, and Supplementary Figure 2.

6. I also don't feel you've satisfied issues around jargon that the other reviewer and I raised. "Hot embedding" (lines 229-231) is unlikely to be familiar to readers of this journal, could you clarify this term? The SAGE/Shipleigh methods used for identifying key features is also not well described. Since these were first published within the last 5 years, it's possible that even experts in feature selection may not be familiar with these methods.

Thank you for this comment. We attempted to address the issue around jargon once again. For example, instead of using the term one-hot encoding to explain how proximities are calculated we simplified the explanation that the aim is to create a binary matrix where each row is a sample and each column the label of the terminal leaf (Lines 339-395). We also reduced our use of terms which are field specific (such as manifold) as much as possible. Finally, we provided a brief explanation on how Shapley values are calculated (Lines 455-498) and removed the calculation of SAGE values to simplify the reporting of results and the analysis. We hope that changes such as this addresses the concerns of both reviewers. We also included Supplementary Figure 1 to provide a visual as to how TreeOrdination creates each projection.

Other Suggestions

1. The mixing and introduction of ASV/OTU seems like a bit of a red herring. My recommendation would be to describe everything in the table as a "feature" since it could potentially be an ASV, OTU, species from metagenomic sequencing, or genome. The method should ultimately be agnostic to the type of feature, it just wants features

We made major modifications in the language throughout the manuscript in an attempt to be more consistent in our language around features, and that ASVs, OTUs, are features and should be considered as such. We hope that these changes improve the clarity of the manuscript and convey to others that this approach can be agnostic to the type of features.

2. Please cite your tools. I shouldn't have to look at your figures to know this was a python implementation or go through your code to figure out the library defaults to know what was in your methods. This is especially important since the scipy Jaccard implementation is slightly different than Jaccard implementations in R.

Also because citation is a very cheap way to keep the resources we all rely on going, since citation shows a compelling need to funders for maintenance

Thank you for this comment. We re-wrote the methods section and did a better job at describing which packages were used and the versions of the packages used.

3. Please check the text sizes on the final figures since most are difficult read at the current DPI. Also, if possible, please provide in-figure legends

New figures were generated. We hope that these are clearer and more informative than previously used ones.

4. Additional remaining issues:

• The scikit libraries are canonically lowercased. (scikit-learn and scikit-bio).

Addressed.

• **The text in several figures is still too small to read at scale in what will probably be the displayed figure. This is a problem in figure S1, figure 5, and several others.**

New figures were generated to hopefully address this problem.

• **You might consider long-transforming p-values when displaying them in boxplots, or possibly presenting them as a swarm plot.**

Addressed (Figure 1)

• **Permutative p-values should always be reported with the number of permutations**

Addressed (Figure 4, Supplementary Figure 1)

References

1. Mentch, L. & Zhou, S. Randomization as Regularization: A Degrees of Freedom Explanation for Random Forest Success. *Journal of Machine Learning Research* **21**, 171:1-171:36 (2020).
2. Rudar, J., Porter, T. M., Wright, M., Golding, G. B. & Hajibabaei, M. LANDMark: An ensemble approach to the supervised selection of biomarkers in high-throughput sequencing data. *BMC Bioinformatics* **23**, 110 (2022).
3. Kuncheva, L. A Bound on Kappa-Error Diagrams for Analysis of Classifier Ensembles. *IEEE Transactions on Knowledge and Data Engineering* **25**, 494–501 (2013).
4. Kuncheva, L. I. & Rodriguez, J. J. Classifier ensembles with a random linear oracle. *IEEE Transactions on Knowledge and Data Engineering* **19**, 500–508 (2007).
5. Suárez-Díaz, J. L., García, S. & Herrera, F. A Tutorial on Distance Metric Learning: Mathematical Foundations, Algorithms, Experimental Analysis, Prospects and Challenges (with Appendices on Mathematical Background and Detailed Algorithms Explanation). (2018)
doi:10.48550/ARXIV.1812.05944.
6. Xiong, C., Johnson, D., Xu, R. & Corso, J. J. Random Forests for Metric Learning with Implicit Pairwise Position Dependence. in *Proceedings of the 18th ACM SIGKDD International Conference on Knowledge Discovery and Data Mining* 958–966 (Association for Computing Machinery, 2012).
doi:10.1145/2339530.2339680.

7. Rhodes, J. S., Cutler, A. & Moon, K. R. Geometry- and Accuracy-Preserving Random Forest Proximities. (2022) doi:10.48550/ARXIV.2201.12682.
8. Martino, C. *et al.* A Novel Sparse Compositional Technique Reveals Microbial Perturbations. *mSystems* **4**, (2019).
9. Anderson, M. J. & Walsh, D. C. I. PERMANOVA, ANOSIM, and the Mantel test in the face of heterogeneous dispersions: What null hypothesis are you testing? *Ecological Monographs* **83**, 557–574 (2013).
10. Martín-Fernández, J. A., Barceló-Vidal, C. & Pawlowsky-Glahn, V. Dealing with Zeros and Missing Values in Compositional Data Sets Using Nonparametric Imputation. *Mathematical Geology* **35**, 253–278 (2003).
11. McMurdie, P. J. & Holmes, S. Waste Not, Want Not: Why Rarefying Microbiome Data Is Inadmissible. *PLoS Computational Biology* **10**, 1003531 (2014).
12. Weiss, S. *et al.* Normalization and microbial differential abundance strategies depend upon data characteristics. *Microbiome* **5**, (2017).
13. Shen, W. X., Liang, S. R., Jiang, Y. Y. & Chen, Y. Z. Enhanced metagenomic deep learning for disease prediction and consistent signature recognition by restructured microbiome 2D representations. *Patterns* **4**, 100658 (2023).
14. Bouranis, J. A. *et al.* Interplay between Cruciferous Vegetables and the Gut Microbiome: A Multi-Omic Approach. *Nutrients* **15**, (2023).
15. Wang, H. *et al.* Perturbed gut microbiome and fecal and serum metabolomes are associated with chronic kidney disease severity. *Microbiome* **11**, 3 (2023).
16. Liu, H. *et al.* MNNMDA: Predicting Human Microbe-Disease Association via a Method to Minimize Matrix Nuclear Norm. *Computational and Structural Biotechnology Journal* (2023) doi:<https://doi.org/10.1016/j.csbj.2022.12.053>.
17. Xi, C. *et al.* Brain-gut microbiota multimodal predictive model in patients with bipolar depression. *Journal of Affective Disorders* **323**, 140–152 (2023).

18. Zhang, H. *et al.* Tracing human life trajectory using gut microbial communities by context-aware deep learning. *Briefings in Bioinformatics* (2023) doi:10.1093/bib/bbac629.
19. Alhusain, L. & Hafez, A. M. Cluster ensemble based on Random Forests for genetic data. *BioData Mining* **10**, 37 (2017).
20. Dalleau, K., Couceiro, M. & Smail-Tabbone, M. Unsupervised Extremely Randomized Trees. in *Advances in Knowledge Discovery and Data Mining* (eds. Phung, D. et al.) 478–489 (Springer International Publishing, 2018).
21. Bernard, S., Cao, H., Sabourin, R. & Heutte, L. Random Forest for Dissimilarity-based Multi-view Learning. in *Handbook of Pattern Recognition and Computer Vision* 119–138 (WORLD SCIENTIFIC, 2020). doi:10.1142/9789811211072_0007.

February 11, 2023

Dr. Josip Rudar
University of Guelph
Integrative Biology
Guelph, Ontario
Canada

Re: Spectrum02065-22R2 (Decision Tree Ensembles Utilizing Multivariate Splits Are Effective at Investigating Beta-Diversity in Medically Relevant 16S Amplicon Sequencing Data)

Dear Dr. Josip Rudar:

Congratulations on the acceptance of your manuscript for publication in Spectrum.

Your manuscript has been accepted, and I am forwarding it to the ASM Journals Department for publication. You will be notified when your proofs are ready to be viewed.

Sincerely,

Jan Claesen
Editor, Microbiology Spectrum
